# Mechanism underlying the DNA-binding preferences of the *Vibrio cholerae* and vibriophage VP882 VqmA quorum-sensing receptors

Olivia P. Duddy[1◉], Xiuliang Huang[1,2◉], Justin E. Silpe[1¤], Bonnie L. Bassler[1,2]*

1 Department of Molecular Biology, Princeton University, Princeton, New Jersey, United States of America,
2 Howard Hughes Medical Institute, Chevy Chase, Maryland, United States of America

◉ These authors contributed equally to this work.
¤ Current address: Department of Chemistry and Chemical Biology, Harvard University, Cambridge, Massachusetts, United States of America
* bbassler@princeton.edu

**Data Availability Statement:** All relevant data are within the manuscript and its Supporting Information files.

## Abstract

Quorum sensing is a chemical communication process that bacteria use to coordinate group behaviors. In the global pathogen *Vibrio cholerae*, one quorum-sensing receptor and transcription factor, called VqmA (VqmA$_{Vc}$), activates expression of the *vqmR* gene encoding the small regulatory RNA VqmR, which represses genes involved in virulence and biofilm formation. Vibriophage VP882 encodes a VqmA homolog called VqmA$_{Phage}$ that activates transcription of the phage gene *qtip*, and Qtip launches the phage lytic program. Curiously, VqmA$_{Phage}$ can activate *vqmR* expression but VqmA$_{Vc}$ cannot activate expression of *qtip*. Here, we investigate the mechanism underlying this asymmetry. We find that promoter selectivity is driven by each VqmA DNA-binding domain and key DNA sequences in the *vqmR* and *qtip* promoters are required to maintain specificity. A protein sequence-guided mutagenesis approach revealed that the residue E194 of VqmA$_{Phage}$ and A192, the equivalent residue in VqmA$_{Vc}$, in the helix-turn-helix motifs contribute to promoter-binding specificity. A genetic screen to identify VqmA$_{Phage}$ mutants that are incapable of binding the *qtip* promoter but maintain binding to the *vqmR* promoter delivered additional VqmA$_{Phage}$ residues located immediately C-terminal to the helix-turn-helix motif as required for binding the *qtip* promoter. Surprisingly, these residues are conserved between VqmA$_{Phage}$ and VqmA$_{Vc}$. A second, targeted genetic screen revealed a region located in the VqmA$_{Vc}$ DNA-binding domain that is necessary to prevent VqmA$_{Vc}$ from binding the *qtip* promoter, thus restricting DNA binding to the *vqmR* promoter. We propose that the VqmA$_{Vc}$ helix-turn-helix motif and the C-terminal flanking residues function together to prohibit VqmA$_{Vc}$ from binding the *qtip* promoter.

**Funding:** This work was supported by the Howard Hughes Medical Institute, National Institutes of Health Grant R37GM065859, and National Science Foundation Grant MCB-1713731 (BLB), NIGMS T32GM007388 (OPD), a Charlotte Elizabeth Procter Fellowship provided by Princeton University, and a National Defense Science and Engineering Graduate Fellowship supported by the Department of Defense (JES). The content is solely the responsibility of the authors and does not necessarily represent the official views of the National Institutes of Health. The funders had no role in study design, data collection and analysis, decision to publish, or preparation of the manuscript.

**Competing interests:** The authors have declared that no competing interests exist.

## Author summary

Bacteria use a chemical communication process called quorum sensing (QS) to orchestrate collective behaviors. Recent studies demonstrate that bacteria-infecting viruses, called phages, also employ chemical communication to regulate collective activities. Phages can encode virus-specific QS-like systems, or they can harbor genes encoding QS components resembling those of bacteria. The latter arrangement suggests the potential for chemical communication across domains, i.e., between bacteria and phages. Ramifications stemming from such cross-domain communication are not understood. Phage VP882 infects the global pathogen *Vibrio cholerae*, and "eavesdrops" on *V. cholerae* QS to optimize the timing of its transition from existing as a parasite to killing the host, and moreover, to manipulate *V. cholerae* biology. To accomplish these feats, phage VP882 relies on $VqmA_{Phage}$, the phage-encoded homolog of the *V. cholerae* $VqmA_{Vc}$ QS receptor and transcription factor. $VqmA_{Vc}$, by contrast, is constrained to the control of only *V. cholerae* genes and is incapable of regulating phage biology. Here, we discover the molecular mechanism underpinning the asymmetric transcriptional preferences of the phage-encoded and bacteria-encoded VqmA proteins. We demonstrate how VqmA transcriptional regulation is crucial to the survival and persistence of both the pathogen *V. cholerae*, and the phage that preys on it.

## Introduction

Quorum sensing (QS) is a cell-cell communication process that allows bacteria to coordinate collective behaviors [1]. QS relies on the production, release, and group-wide detection of extracellular signaling molecules called autoinducers (AIs). In the global pathogen *Vibrio cholerae*, the AI, 3,5-dimethyl-pyrazin-2-ol (DPO), together with its partner cytoplasmic QS receptor and transcription factor, VqmA ($VqmA_{Vc}$), comprises one of the QS circuits that controls group behaviors [2–4]. $VqmA_{Vc}$, following binding to DPO, activates transcription of the *vqmR* gene encoding the small RNA, VqmR, which, in turn, represses the expression of genes required for biofilm formation and virulence factor production [2–4].

Recently, bacteria-specific viruses, called phages, have been shown to engage in density-dependent regulation of their lysis-lysogeny decisions via chemical dialogs [5,6]. Germane to our studies are phages that encode proteins resembling bacterial QS components [5,7]. Vibriophage VP882 is one such phage: It encodes the QS receptor VqmA ($VqmA_{Phage}$), a homolog of the *V. cholerae* QS receptor $VqmA_{Vc}$ [5]. $VqmA_{Phage}$, like $VqmA_{Vc}$, binds host-produced DPO. DPO-bound $VqmA_{Phage}$ activates transcription of the phage gene *qtip*. Qtip is an antirepressor that sequesters the phage VP882 repressor of lysis, leading to derepression of the phage lytic program and killing of the *Vibrio* host at high cell density [5,8]. Thus, the DPO AI mediates both bacterial and phage lifestyle decisions. Curiously, $VqmA_{Phage}$ can substitute for $VqmA_{Vc}$ to activate the *V. cholerae vqmR* promoter (P*vqmR*) [5]. In contrast, $VqmA_{Vc}$ cannot substitute for $VqmA_{Phage}$ and recognize the phage VP882 *qtip* promoter (P*qtip*). Presumably, the ability of $VqmA_{Phage}$ to bind both P*vqmR* and P*qtip* provides phage VP882 the capacity to influence host QS and simultaneously enact its own lysis-lysogeny decision.

$VqmA_{Phage}$ shares ~43% amino acid sequence identity with $VqmA_{Vc}$, and most of the key residues required for ligand and DNA binding are conserved [5,9]. Thus, how $VqmA_{Phage}$ can recognize two different promoters, while $VqmA_{Vc}$ cannot, is not understood. Here, we define the mechanism underlying this asymmetry. We show that VqmA selectivity for target promoters is driven by the DNA-binding domain (DBD) of the respective protein. We identify 6 key

nucleotides within P*vqmR* and P*qtip* that contribute to VqmA promoter-binding selectivity, as exchanging these critical DNA sequences inverts the DNA-binding preferences of the two VqmA proteins. The 192$^{nd}$ and 194$^{th}$ residues in VqmA$_{Vc}$ and VqmA$_{Phage}$, respectively, within the helix-turn-helix (HTH) motifs, contribute to promoter-binding specificity. Isolation of VqmA$_{Phage}$ mutants capable of activating *vqmR* expression but incapable of activating *qtip* expression revealed conserved or functionally conserved residues in VqmA$_{Phage}$ and VqmA$_{Vc}$, indicating that VqmA$_{Vc}$ likely possesses an additional feature that prevents it from binding P*qtip* DNA. A mosaic VqmA$_{Vc}$ protein containing the VqmA$_{Phage}$ HTH motif along with the C-terminal 25 flanking VqmA$_{Phage}$ residues was capable of binding P*qtip*. Thus, the two corresponding regions in VqmA$_{Vc}$ must function in concert to prevent VqmA$_{Vc}$ from binding to P*qtip*. Together, our analyses demonstrate how VqmA$_{Phage}$, via its promiscuous DNA-binding activity, can control phage VP882 functions and drive host *V. cholerae* QS. Moreover, we discover why *V. cholerae* VqmA$_{Vc}$ cannot do the reverse, as its DNA binding is strictly constrained to the host *V. cholerae* genome.

## Results

### VqmA promoter-binding selectivity is conferred by the DNA-binding domain

VqmA proteins are composed of N-terminal Per-Arnt-Sim (PAS) domains responsible for binding the DPO AI and C-terminal DBDs containing HTH motifs [10]. Both VqmA$_{Vc}$ and VqmA$_{Phage}$ bind DPO. By contrast, with respect to DNA binding, VqmA$_{Phage}$ binds to P*qtip* and P*vqmR*, whereas VqmA$_{Vc}$ only binds to P*vqmR* [5]. We reasoned that this asymmetric DNA-binding pattern arises from differences in the DBDs (S1 Fig). To test this idea, we constructed chimeras in which we exchanged the VqmA$_{Vc}$ and VqmA$_{Phage}$ C-terminal domains to produce $_{Vc}$N-C$_{Phage}$ and $_{Phage}$N-C$_{Vc}$ proteins. We chose to make the junction at a residue near the C-terminal end of the PAS domain immediately following an amino acid stretch (GTIF) that is identical in both VqmA$_{Vc}$ and VqmA$_{Phage}$ (S1 Fig). We cloned *vqmA*$_{Vc}$, *vqmA*$_{Phage}$, $_{Vc}$N-C$_{Phage}$, and $_{Phage}$N-C$_{Vc}$ under an arabinose-inducible promoter and transformed each construct into recombinant Δ*tdh E. coli* harboring a P*vqmR-lux* or a P*qtip-lux* reporter. The Tdh enzyme is required for DPO biosynthesis, therefore a Δ*tdh E. coli* strain makes no DPO [3]. Apo-VqmA displays basal transcriptional activity *in vivo* [9]. Thus, while DPO enhances VqmA DNA-binding activity, it is not an absolute requirement for binding. Using Δ*tdh E. coli* for these studies ensured that any transcriptional activity that occurred was exclusively a consequence of the DNA-binding capabilities of the chimeras and not ligand-binding-driven transcriptional activation of the chimeras. Consistent with our hypothesis, promoter activation by each chimera was determined by the protein from which the DBD originated: All four versions of VqmA activated P*vqmR-lux*, whereas only VqmA$_{Phage}$ and $_{Vc}$N-C$_{Phage}$ activated P*qtip-lux* (Fig 1A and 1B, respectively). Next, we conjugated the four versions of VqmA into Δ*tdh* Δ*vqmA*$_{Vc}$ *V. cholerae* lysogenized by a phage VP882 mutant in which the endogenous *vqmA*$_{Phage}$ was inactive (VP882 *vqmA*$_{Phage}$::Tn5). Thus, the only source of VqmA protein was that made from the plasmid. As expected, following arabinose-induction, only VqmA$_{Phage}$ and $_{Vc}$N-C$_{Phage}$ activated *qtip* expression and induced host-cell lysis (Fig 1C).

We verified the above findings *in vitro* using electrophoretic mobility shift assays (EMSAs). Consistent with the cell-based assays, the purified VqmA$_{Vc}$, VqmA$_{Phage}$, $_{Vc}$N-C$_{Phage}$, and $_{Phage}$N-C$_{Vc}$ proteins shifted P*vqmR* DNA, whereas only the VqmA$_{Phage}$ and $_{Vc}$N-C$_{Phage}$ proteins shifted P*qtip* DNA (Fig 1D). Assessing the ratios of bound to total DNA across varying protein concentrations allowed us to calculate the relative binding affinities (EC$_{50}$) of the VqmA proteins for P*vqmR* and P*qtip* DNA (S2A Fig). Our EMSA analyses show that

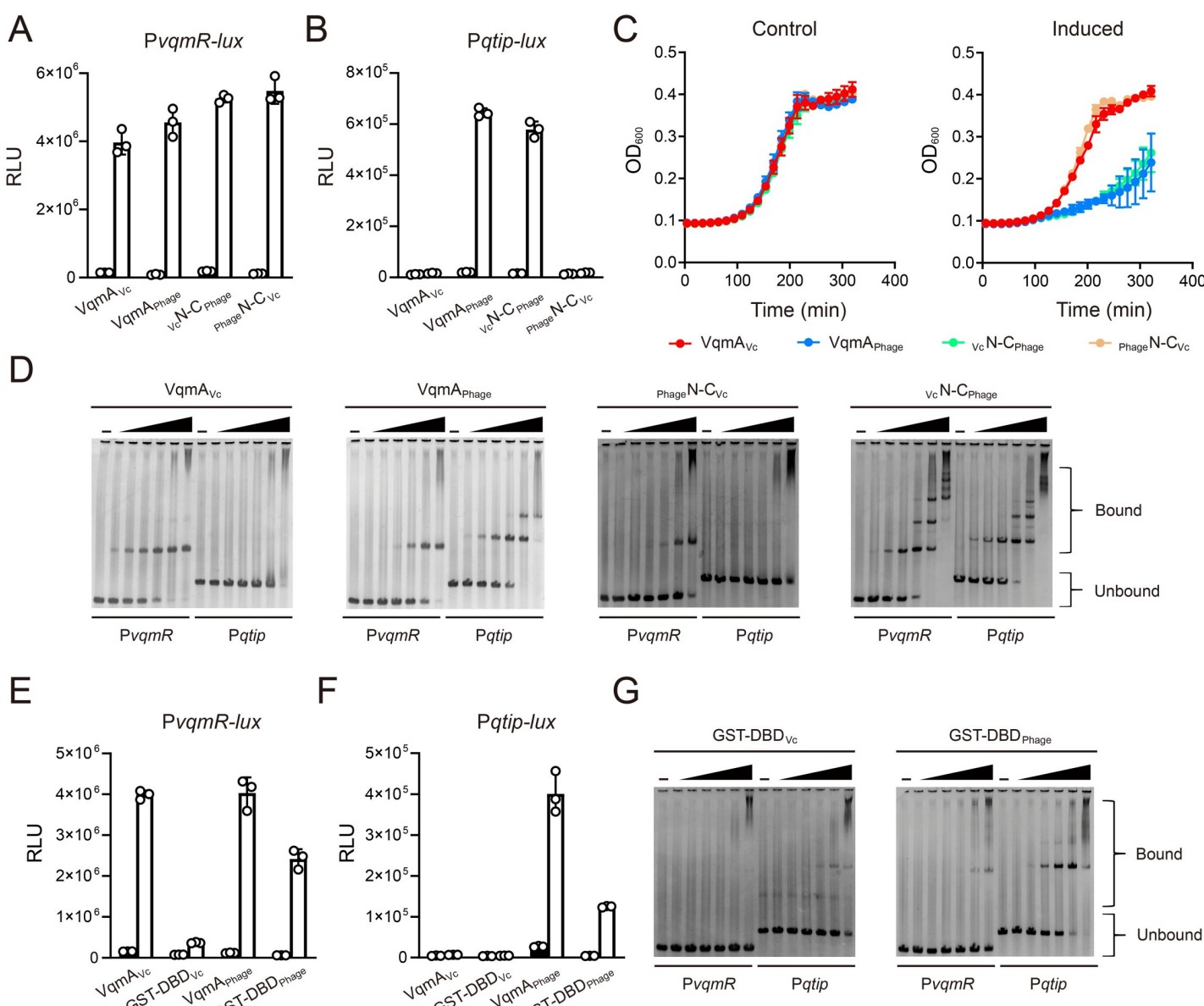

**Fig 1. Promoter DNA-binding selectivity is conferred by the VqmA DBD.** (A and B) Normalized reporter activity from Δ*tdh E. coli* harboring (A) P*vqmR-lux* or (B) P*qtip-lux* and arabinose-inducible VqmA$_{Vc}$, VqmA$_{Phage}$, $_{Vc}$N-C$_{Phage}$, or $_{Phage}$N-C$_{Vc}$. Black, no arabinose; white, 0.2% arabinose. Data are represented as mean ± SD (error bars) with *n* = 3 biological replicates. (C) Growth curves of the Δ*tdh* Δ*vqmA$_{Vc}$ V. cholerae* harboring phage VP882 *vqmA$_{Phage}$*::Tn5 and arabinose-inducible VqmA$_{Vc}$, VqmA$_{Phage}$, $_{Vc}$N-C$_{Phage}$, or $_{Phage}$N-C$_{Vc}$ in medium lacking (Control) or containing 0.2% arabinose (Induced). (D) EMSAs showing binding of VqmA proteins to P*vqmR* and P*qtip* DNA. From left to right are, VqmA$_{Vc}$, VqmA$_{Phage}$, $_{Phage}$N-C$_{Vc}$, and $_{Vc}$N-C$_{Phage}$. 25 nM P*vqmR* or P*qtip* DNA was used in all EMSAs with no protein (designated -) or 2-fold serial dilutions of proteins. The lowest and highest protein (dimer) concentrations are 18.75 nM and 600 nM, respectively. (E and F) Normalized reporter activity from WT *E. coli* as in panels A and B harboring arabinose-inducible VqmA$_{Vc}$, GST-DBD$_{Vc}$, VqmA$_{Phage}$, and GST-DBD$_{Phage}$. (G) EMSAs showing binding of GST-DBD$_{Vc}$ and GST-DBD$_{Phage}$ to P*vqmR* and P*qtip* DNA. Probe and protein concentrations as in panel D.

$_{Phage}$N-C$_{Vc}$, like VqmA$_{Vc}$, only bound P*vqmR*, but with an estimated ~7-fold lower affinity. Consistent with our previous findings, VqmA$_{Phage}$ bound P*qtip* about 3-fold more strongly than it bound P*vqmR* [5]. By contrast, $_{Vc}$N-C$_{Phage}$ showed a modest increase in its preference for P*qtip* relative to that for P*vqmR*, with binding to both promoters at a level similar to that with which VqmA$_{Phage}$ bound P*qtip*. Indeed, in agreement with our EC$_{50}$ measurements, when P*qtip* and P*vqmR* DNA were supplied at equimolar concentrations in a competitive

DNA-binding assay, lower amounts of VqmA$_{Phage}$ and $_{Vc}$N-C$_{Phage}$ were required to shift P*qtip* DNA than to shift P*vqmR* DNA (S2B Fig). In conclusion and in agreement with our *in vivo* results, the respective DBD of each purified VqmA protein drives promoter selectively.

We next assayed the VqmA$_{Vc}$ and VqmA$_{Phage}$ DBDs lacking their PAS domains (DBD$_{Vc}$ and DBD$_{Phage}$, respectively) for activation of P*vqmR-lux* and P*qtip-lux*. Deletion of the PAS domains resulted in inactive proteins as neither DBD activated transcription (S3A and S3B Fig, respectively), and likewise, EMSA analyses showed that neither DBD bound either promoter (S3C Fig). Gel filtration analyses indicated that the DBD proteins purified as monomers (S3D Fig), suggesting that the DBDs were unable to dimerize in the absence of their partner PAS domains. This result is consistent with previous findings that, in addition to sensing DPO, the VqmA$_{Vc}$ PAS domain is responsible for dimerization [9,11].

Transcriptional activity driven by HTH-containing proteins typically depends on dimer formation. Soluble glutathione S-transferase (GST) spontaneously forms a homodimer [12], and so GST can be employed as a substitute for native dimerization domains of proteins [13]. Thus, to examine the VqmA requirement for dimerization, we fused GST to the N-terminus of each VqmA DBD to yield recombinant GST-DBD$_{Vc}$ and GST-DBD$_{Phage}$ and we tested whether DNA-binding function was restored. Indeed, the GST-DBD proteins purified as dimers (S3D Fig). P*vqmR-lux* and P*qtip-lux* expression analyses revealed that the DBDs, when fused to GST, regained function, with the caveat that the GST-DBD$_{Vc}$ exhibited 10-fold reduced activity compared to wild-type (WT) VqmA$_{Vc}$ (Fig 1E). Importantly, the DNA-binding preferences mimicked those of the full-length proteins: GST-DBD$_{Phage}$ activated both P*vqmR-lux* and P*qtip-lux*, whereas GST-DBD$_{Vc}$ only activated P*vqmR-lux* (Fig 1E and 1F). Companion EMSA analyses showed that GST-DBD$_{Phage}$ bound P*qtip* ~5-fold more strongly than it bound P*vqmR*, whereas GST-DBD$_{Vc}$ showed almost no binding to P*vqmR* and, unexpectedly, some weak binding could be detected to the P*qtip* DNA (Fig 1G). We confirmed that purified GST alone did not bind either P*vqmR* or P*qtip* (S3E Fig). Given that the GST-DBD$_{Vc}$ driven activation of P*qtip-lux* was undetectable *in vivo* (Fig 1F), we presume that the observed *in vitro* GST-DBD$_{Vc}$ binding to P*qtip* DNA is a consequence of the simplified context in which the EMSA is performed. Likely, the DNA:VqmA ratio in the EMSA is far higher than in cells, which, in the case of GST-DBD$_{Vc}$, fosters modest non-specific DNA binding. Taken together, our results show that VqmA promoter-binding selectivity is conferred by the DBD, and that dimerization is necessary.

## VqmA DNA-binding preferences can be inverted by exchanging key DNA sequences in P*vqmR* and P*qtip*

To study the VqmA promoter-binding asymmetry from the aspect of the DNA, our next goal was to identify the critical DNA sequence within P*qtip* that prevents VqmA$_{Vc}$ from binding. In the phage VP882 genome, P*qtip* resides between *vqmA$_{Phage}$* and *qtip* and VqmA$_{Phage}$ activates its own and *qtip* expression, suggesting that VqmA$_{Phage}$ binding may involve both DNA strands. Similarly, VqmA$_{Vc}$ has been shown to interact with both strands of P*vqmR* [11]. Thus, in each case, both DNA strands need to be considered (Fig 2A). Previous work revealed that the critical region in P*vqmR* required for VqmA$_{Vc}$ binding is -AGGGGGGATTTCCCCCCT- [2,11]. The corresponding fragment from P*qtip*, but on the opposite DNA strand, -TAGGGG GAAAAATACCCT-, possesses ~56% sequence identity to this region suggesting it could be the key stretch of DNA that drives VqmA$_{Phage}$ promoter selection. The highest divergence in the two promoters is in the central 6 nucleotides: "-AAAATA-" in P*qtip* and "-TTTCCC-" in P*vqmR*. We synthesized DNA probes in which we exchanged the "-AAAATA-" in P*qtip* with "-TTTCCC-" from P*vqmR* and tested VqmA$_{Vc}$ and VqmA$_{Phage}$ binding by EMSA analysis.

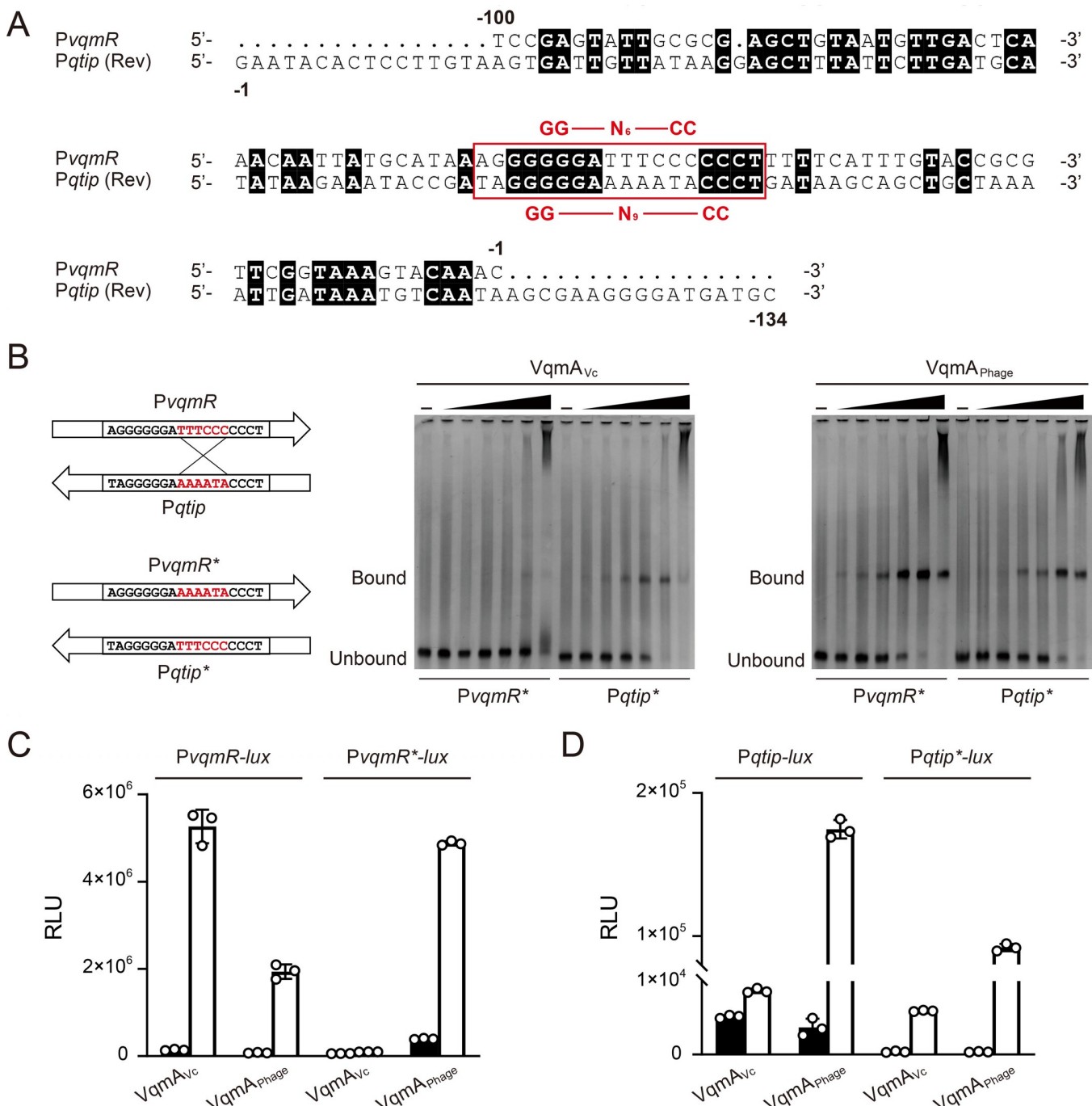

**Fig 2. Promoter selectivity is reversed by exchanging key nucleotide fragments.** (A) DNA sequence alignment (ClustalW) of P*vqmR* and P*qtip*. The reverse strand of P*qtip* is shown. Numbering indicates positions relative to the transcription start sites. Identical nucleotides are designated with black shading. The reported 18-bp DNA stretch in P*vqmR* required for VqmA$_{Vc}$ to bind (2,11) and the corresponding region in P*qtip* are highlighted in the red box. The GG-N$_6$-CC palindrome in P*vqmR* (2,11) and the recently identified GG-N$_9$-CC palindrome in P*qtip* (15) are indicated above and below the red box, respectively. (B) EMSAs showing binding of the designated VqmA proteins to P*vqmR\** and P*qtip\** DNA. The cartoon at the left illustrates the key sequences exchanged in the probes. Probe and protein concentrations as in Fig 1D. (C) Normalized reporter activity from Δ*tdh E. coli* harboring P*vqmR-lux* or P*vqmR\*-lux* and arabinose-inducible VqmA$_{Vc}$ or VqmA$_{Phage}$. Black, no arabinose; white, 0.2% arabinose. Data are represented as mean ± SD (error bars) with *n = 3* biological replicates. (D) As in C for P*qtip-lux* or P*qtip\*-lux*.

We call these probes P*vqmR** and P*qtip**, respectively. Indeed, promoter DNA-binding specificity was exchanged: $VqmA_{Vc}$ shifted P*qtip**, whereas it only weakly shifted P*vqmR** (Fig 2B). $VqmA_{Phage}$ bound to P*vqmR** twice as strongly as it bound to P*qtip**, showing the opposite preference for the two synthetic promoters compared to the native promoters (Fig 2B). P*vqmR**-*lux* and P*qtip**-*lux* transcriptional fusions mimicked the EMSA results: $VqmA_{Vc}$ only activated expression of P*qtip**-*lux*, whereas $VqmA_{Phage}$ activated expression of P*vqmR**-*lux* and P*qtip**-*lux* (Fig 2C and 2D). Thus, this 6-nucleotide stretch is the key sequence that determines the DNA-binding specificity for the two VqmA proteins. Moreover, the presence of the -AAAATA- nucleotide sequence in P*qtip* is sufficient to prevent $VqmA_{Vc}$ from activating transcription of P*qtip*.

## Protein sequence-guided mutagenesis reveals that residue E194 in phage VP882 $VqmA_{Phage}$ and the equivalent A192 residue in *V. cholerae* $VqmA_{Vc}$ contribute to specificity for P*qtip*

We considered two possible mechanisms that could underpin the asymmetric VqmA DNA-binding patterns: phage VP882 $VqmA_{Phage}$ could possess a feature that relaxes its DNA-binding specificity, and/or *V. cholerae* $VqmA_{Vc}$ could possess a feature that restricts its DNA-binding ability. To distinguish between these possibilities, we first probed which residues drive $VqmA_{Phage}$ interactions with P*qtip* but do not contribute to interactions with P*vqmR*. To do this, we performed site-directed mutagenesis of $VqmA_{Phage}$ with the goal of identifying mutants that fail to bind P*qtip* but retain binding to P*vqmR*. Charged residues in HTH motifs typically mediate interactions between VqmA-type transcription factors and DNA, and indeed, both VqmA HTHs are enriched in positively-charged amino acids [9,11,14]. Sequence alignment of the HTHs in $VqmA_{Phage}$ and $VqmA_{Vc}$ revealed four obvious differences in charged residues that could underlie the DNA-binding asymmetry between the two proteins (S1 Fig). We mutated those residues in $VqmA_{Phage}$ to the corresponding $VqmA_{Vc}$ residues. The changes are: $VqmA_{Phage}^{K176Q}$, $VqmA_{Phage}^{R184I}$, $VqmA_{Phage}^{I193E}$, and $VqmA_{Phage}^{E194A}$. To test the combined effect of these mutations on $VqmA_{Phage}$ DNA-binding function, we also constructed the quadruple $VqmA_{Phage}^{K176Q, R184I, I193E, E194A}$ mutant. $VqmA_{Phage}^{K176Q}$, $VqmA_{Phage}^{R184I}$, $VqmA_{Phage}^{I193E}$ retained the ability to induce phage lysis showing that *in vivo* binding to P*qtip* was not eliminated (Fig 3A). $VqmA_{Phage}^{E194A}$ induced only low-level cell lysis suggesting that, while binding to P*qtip* is not eliminated, it is compromised (Fig 3A). Analysis of P*vqmR-lux* and P*qtip-lux* expression revealed that all four $VqmA_{Phage}$ single point mutants possessed levels of activity within 2-fold of that of WT P*vqmR-lux*. By contrast, they displayed ~2-7-fold reductions in P*qtip-lux* activity, with $VqmA_{Phage}^{E194A}$ being the least active (Fig 3B and 3C, respectively). The quadruple mutant was unable to induce phage lysis in a *V. cholerae* lysogen and it did not activate P*vqmR-lux* or P*qtip-lux* expression showing it is defective in binding to both promoters (Fig 3A–3C). Western blot analysis demonstrated that all of the $VqmA_{Phage}$ variants were produced at levels similar to WT in both *V. cholerae* and *E. coli* (S4A Fig). Thus, our results indicate that, among these charged residues, only the $VqmA_{Phage}$ residue E194 in the HTH motif plays a role in $VqmA_{Phage}$ selection of P*qtip*.

While the residues we mutated in the phage VP882 $VqmA_{Phage}$ HTH motif do not dramatically perturb site-specific recognition of P*qtip*, the corresponding residues in the *V. cholerae* $VqmA_{Vc}$ HTH motif could nonetheless restrict its capacity to bind P*qtip*. Therefore, we also mutated the analogous $VqmA_{Vc}$ residues to the corresponding $VqmA_{Phage}$ residues. We made: $VqmA_{Vc}^{Q174K}$, $VqmA_{Vc}^{I182R}$, $VqmA_{Vc}^{E191I}$, $VqmA_{Vc}^{A192E}$, and $VqmA_{Vc}^{Q174K, I182R, E191I, A192E}$. Here, our goal was to test whether the variants gained the ability to bind P*qtip*. Only $VqmA_{Vc}^{A192E}$ induced a modest level of lysis in the *V. cholerae* lysogen, whereas all other

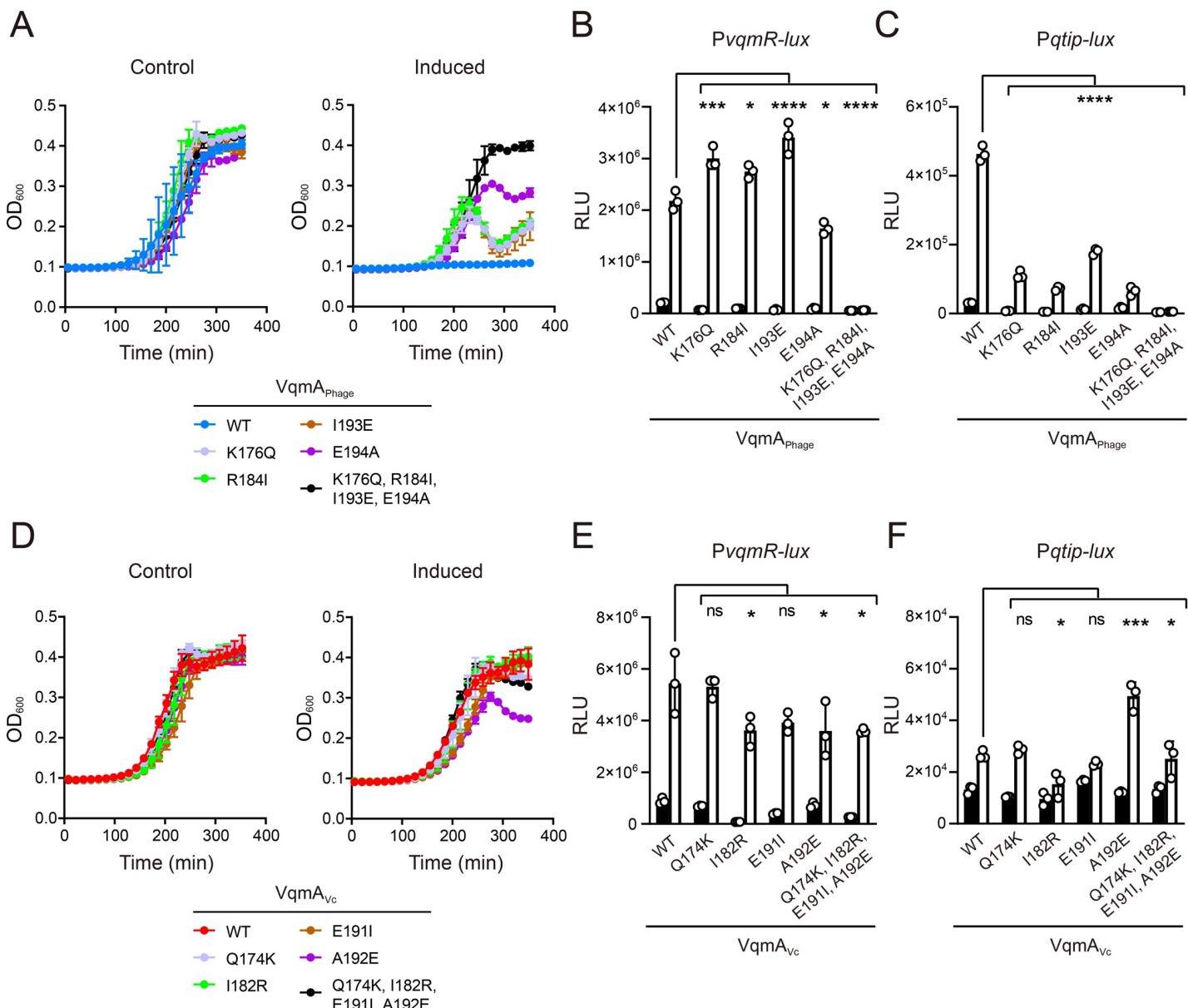

**Fig 3. VqmA_Phage residue E194 and the corresponding VqmA_Vc residue A192 contribute to specificity for binding to P*qtip*.** (A) Growth curves of Δ*tdh* Δ*vqmA_Vc* *V. cholerae* harboring phage VP882 *vqmA_Phage*::Tn*5* and the indicated 3xFLAG-VqmA_Phage alleles in medium lacking (Control) or containing 0.2% arabinose (Induced). (B and C) Normalized reporter activity from Δ*tdh E. coli* harboring (B) P*vqmR-lux* or (C) P*qtip-lux* and the indicated arabinose-inducible 3xFLAG-VqmA_Phage alleles. Black, no arabinose; white, 0.2% arabinose. Data are represented as mean ± SD (error bars) with *n* = 3 biological replicates. (D) Growth curves of Δ*tdh* Δ*vqmA_Vc* *V. cholerae* harboring phage VP882 *vqmA_Phage*::Tn*5* and the indicated 3xFLAG-VqmA_Vc alleles in medium lacking (Control) or containing 0.2% arabinose (Induced). (E and F) Normalized reporter activity from Δ*tdh E. coli* harboring (E) P*vqmR-lux* or (F) P*qtip-lux* and the indicated arabinose-inducible 3xFLAG-VqmA_Vc alleles. Black, no arabinose; white, 0.2% arabinose. Data are represented as mean ± SD (error bars) with *n* = 3 biological replicates. ns = not significant, \*\*\*\**P* <0.0001, \*\*\**P* <0.0005, \*\**P* <0.005, \**P* <0.05 in one-way ANOVA compared to WT VqmA proteins.

VqmA_Vc variants failed to do so (Fig 3D). All of the VqmA_Vc variants drove the WT level of P*vqmR-lux* activity (Fig 3E). VqmA_Vc^A192E generated low but detectable P*qtip-lux* expression, while the other VqmA_Vc variants did not (Fig 3F). The VqmA_Vc variants were produced at similar levels to WT VqmA_Vc in *V. cholerae* and *E. coli* (S4B Fig). We conclude that, among the tested residues, only A192 plays a role in preventing VqmA_Vc from binding P*qtip*.

Our mutagenesis analyses for VqmA$_{Vc}$ are consistent with our analyses for VqmA$_{Phage}$: The residue at the 192$^{nd}$ position in *V. cholerae* VqmA$_{Vc}$ and the analogous residue at the 194$^{th}$ position in phage VP882 VqmA$_{Phage}$ contribute to selection of P*qtip*. However, given that the A192E substitution in VqmA$_{Vc}$ results in only partial activation of P*qtip* expression, and the E194A substitution in VqmA$_{Phage}$ results in only partial loss of activation of P*qtip*, the E194 residue in VqmA$_{Phage}$ cannot be the sole amino acid responsible for the preference VqmA$_{Phage}$ shows for P*qtip*. Rather, additional residues in VqmA$_{Phage}$ must participate in conferring specificity.

## Random mutagenesis of the VqmA$_{Phage}$ DBD reveals that residues G201, A202, E207, and M211 are required for VqmA$_{Phage}$ to bind P*qtip* but are dispensable for binding P*vqmR*

Our protein sequence-guided approach did not reveal the primary mechanism underlying promoter-binding specificity for either of the VqmA proteins. We therefore performed a genetic screen to forward our goal of identifying phage VP882 VqmA$_{Phage}$ mutants that fail to bind P*qtip* but retain the ability to bind P*vqmR*. We constructed a library of random mutations in the region of *vqmA$_{Phage}$* encoding the DBD in the context of the full-length gene, cloned them into a plasmid under an arabinose-inducible promoter, and introduced them into Δ*tdh* Δ*vqmA$_{Vc}$ V. cholerae* harboring P*vqmR-lux* on the chromosome and lysogenized by phage VP882 harboring inactive *vqmA$_{Phage}$* (*vqmA$_{Phage}$*::Tn5). The logic of the screen is as follows: When propagated on agar plates supplemented with arabinose, *V. cholerae* exconjugants harboring *vqmA$_{Phage}$* alleles possessing reasonable P*qtip*-binding activity will lyse because those VqmA$_{Phage}$ proteins will bind P*qtip* on the phage VP882 genome and launch the phage lytic cascade (S5 Fig). Such exconjugants will die and thus be eliminated from the screen. Exconjugants that survive but carry *vqmA$_{Phage}$* null alleles will produce no light because those VqmA$_{Phage}$ proteins will fail to bind P*vqmR-lux*, so they also can be eliminated from the screen. The *vqmA$_{Phage}$* alleles of interest to us are those that are maintained in surviving exconjugants (because they encode proteins that cannot bind P*qtip*) and produce light (because they encode proteins that can bind P*vqmR-lux*).

Our screen yielded the following mutants: VqmA$_{Phage}$$^{G201D}$, VqmA$_{Phage}$$^{G201R}$, VqmA$_{Phage}$$^{A202V}$, VqmA$_{Phage}$$^{E207K}$, VqmA$_{Phage}$$^{E207V}$, and VqmA$_{Phage}$$^{M211K}$ (Fig 4A). To verify that these VqmA$_{Phage}$ mutants were indeed defective in binding P*qtip*, we individually transformed them into Δ*tdh E. coli* carrying the P*qtip-lux* reporter or the P*vqmR-lux* reporter and measured light production. All variants retained WT capability to activate P*vqmR-lux*, but they did not harbor WT capability to activate P*qtip-lux* expression (>10-fold reductions in activity) (Fig 4B and 4C, respectively). Thus, any residual P*qtip* binding by these mutant VqmA$_{Phage}$ proteins is insufficient to induce host-cell lysis in the phage VP882 lysogen (Fig 4A). We verified that the VqmA$_{Phage}$ variants are produced at the same level as WT VqmA$_{Phage}$ in *V. cholerae* and *E. coli* (S4C Fig). According to the protein sequence alignment, VqmA$_{Phage}$ residues (175–200) corresponding to positions 173–198 in VqmA$_{Vc}$ comprise the VqmA$_{Phage}$ HTH motif (S1 Fig). Thus, the residues identified in the mutagenesis (G201, A202, E207, and M211) are located C-terminal to the VqmA$_{Phage}$ HTH motif. Mapping the analogous *V. cholerae* VqmA$_{Vc}$ residues (G199, A200, Q205, and L209) to the DPO-VqmA$_{Vc}$-P*vqmR* structure (there is no DPO-VqmA$_{Phage}$-P*qtip* structure) also shows that all of these residues cluster in a flexible loop region and helix adjacent to, but distinct from the HTH motif that directly contacts DNA (Figs 4D and S1). Surprisingly, the residues identified in the VqmA$_{Phage}$ mutagenesis are either identical (VqmA$_{Phage}$ G201 and A202 versus VqmA$_{Vc}$ G199 and A200) or similar (VqmA$_{Phage}$ E207 and M211 versus VqmA$_{Vc}$ Q205 and L209) between VqmA$_{Phage}$

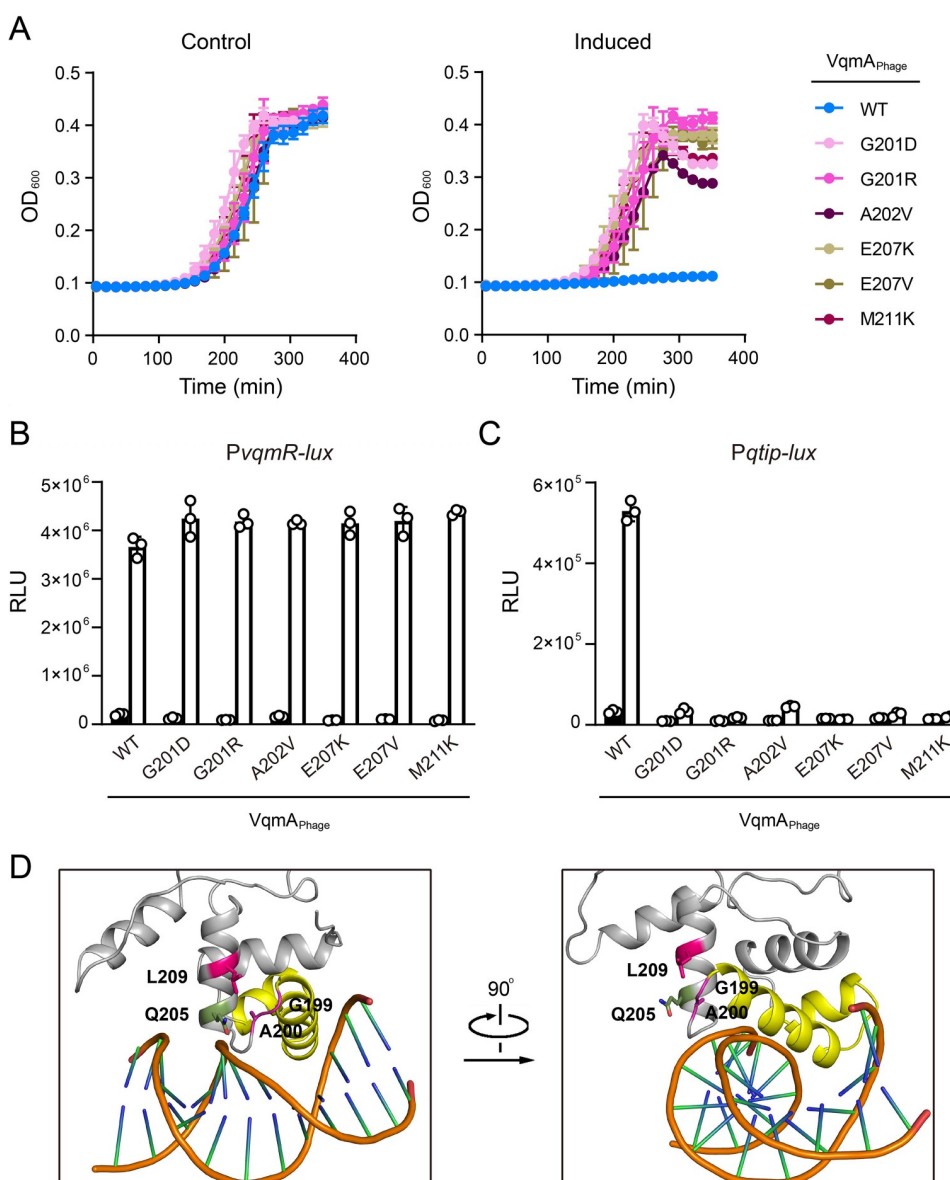

**Fig 4. VqmA_Phage residues G201, A202, E207, and M211 are required for binding to Pqtip.** (A) Growth curves of Δ*tdh* Δ*vqmA_Vc* *V. cholerae* harboring phage VP882 *vqmA_Phage*::Tn5 and the indicated 3xFLAG-VqmA_Phage alleles in medium lacking (Control) or containing 0.2% arabinose (Induced). (B and C) Normalized reporter activity from Δ*tdh* *E. coli* harboring (B) P*vqmR-lux* or (C) P*qtip-lux* and the indicated arabinose-inducible 3xFLAG-VqmA_Phage alleles. Black, no arabinose; white 0.2% arabinose. Data are represented as mean ± SD (error bars) with *n = 3* biological replicates. (D) Close up views of the DBD from the crystal structure of DPO-VqmA_Vc bound to P*vqmR* (PDB: 6ide, protein in gray with the HTH motif in yellow, and the DNA in orange). The color scheme for VqmA_Vc residues G199, A200, Q205, and L209 mirrors that used in panel A.

and VqmA_Vc. To test whether possession of the similar residues is sufficient to confer DNA-binding specificity for P*qtip*, we constructed VqmA_Vc^Q205E and VqmA_Vc^L209M and tested their DNA-binding functions as above. VqmA_Vc^Q205E and VqmA_Vc^L209M, like WT VqmA_Vc, activated P*vqmR-lux* but failed to activate P*qtip-lux* (S6A and S6B Fig, respectively). We make the following four conclusions from these findings: 1) There are at least four residues (G201, A202, E207, and M211) required for VqmA_Phage to recognize P*qtip* DNA. 2) Because the

$VqmA_{Phage}$ G201D, G201R, A202V, E207K, E207V, and M211K variants exhibit WT binding to P*vqmR*, the substitutions at these four residues must not significantly affect P*vqmR* recognition. 3) Because these residues are conserved or similar between $VqmA_{Phage}$ and $VqmA_{Vc}$, one would expect $VqmA_{Vc}$ to have the capacity to bind P*qtip*. 4) However, because $VqmA_{Vc}$ in fact does not bind P*qtip*, $VqmA_{Vc}$ likely possesses an additional feature that resides elsewhere in the protein that prevents P*qtip* binding from occurring.

## The restrictive element that prevents $VqmA_{Vc}$ from binding P*qtip* is located in its HTH motif and the adjacent C-terminal region of 25 residues

To test the hypothesis that a feature in the $VqmA_{Vc}$ DBD restricts its DNA-binding capacity to P*vqmR*, we performed a genetic screen aimed at identifying $VqmA_{Vc}$ mutants capable of activating P*qtip-lux* expression. To do this, we constructed a library of random *vqmA_Vc* DBD alleles containing, on average, 1–2 substitutions, and we cloned them into a plasmid under an arabinose-inducible promoter. The library was transformed into the $\Delta tdh$ *E. coli* strain harboring the P*qtip-lux* reporter and transformants were propagated on plates containing arabinose. We screened ~10,000 transformants for colonies that produced light indicating that they contained $VqmA_{Vc}$ proteins that activated P*qtip-lux*. This strategy yielded no such transformants. Several possibilities could explain our result: We did not screen sufficient numbers of mutants, the mutagenesis did not yield the crucial change, or no alteration of a single residue can enable $VqmA_{Vc}$ binding to P*qtip*.

We expanded our search for the DNA-binding restrictive element present in $VqmA_{Vc}$ by assessing whether a particular region in the $VqmA_{Vc}$ DBD constrains promoter binding to P*vqmR*. To do this, we constructed five $VqmA_{Vc}$ mosaic proteins by replacing ~20–30 residues in the *V. cholerae* $VqmA_{Vc}$ DBD with the corresponding residues from the phage VP882 $VqmA_{Phage}$ DBD. We call these proteins $VqmA_{Vc}{}^{*126-149}$, $VqmA_{Vc}{}^{*150-170}$, $VqmA_{Vc}{}^{*171-199}$, $VqmA_{Vc}{}^{*200-224}$, and $VqmA_{Vc}{}^{*225-246}$ (see S1 Fig for relevant protein segments). Each superscript denotes the $VqmA_{Vc}$ amino acid residues that have been replaced by the corresponding residues from $VqmA_{Phage}$. In all the mosaics, either the intact $VqmA_{Vc}$ HTH or the intact $VqmA_{Phage}$ HTH was present. For reference, the $VqmA_{Vc}$ HTH motif consists of residues 173 to 198 and the $VqmA_{Phage}$ HTH spans residues 175 to 200. We tested the mosaic $VqmA_{Vc}$ proteins for activation of the P*vqmR-lux* and P*qtip-lux* reporters. The DNA specificity of all the $VqmA_{Vc}$ mosaics mimicked WT $VqmA_{Vc}$ as P*vqmR-lux* was expressed but P*qtip-lux* was not (Fig 5A and 5B, respectively). We confirmed that the mosaic $VqmA_{Vc}$ proteins are expressed at levels similar to WT $VqmA_{Vc}$ (S7 Fig). Our results suggest that the feature that prevents *V. cholerae* $VqmA_{Vc}$ from binding to P*qtip* is larger than the regions delineated by any of the $VqmA_{Vc}$ mosaics, or it could be that multiple patches in the $VqmA_{Vc}$ DBD that are not contiguous in amino acid sequence are responsible.

Pinpointing non-contiguous regions that could, together, contain the $VqmA_{Vc}$ restrictive element is challenging. However, testing for a larger contiguous expanse that could contain the putative restrictive element is straightforward. Thus, we constructed two additional *V. cholerae* $VqmA_{Vc}$ mosaic proteins. In one construct, called $VqmA_{Vc}{}^{*150-199}$, we introduced the $VqmA_{Phage}$ HTH along with the immediate N-terminal 25 amino acids in place of the corresponding $VqmA_{Vc}$ region. Second, in a construct called $VqmA_{Vc}{}^{*171-224}$, we introduced the $VqmA_{Phage}$ HTH together with the immediate C-terminal 25 amino acid stretch in place of that $VqmA_{Vc}$ region. $VqmA_{Vc}{}^{*150-199}$ and $VqmA_{Vc}{}^{*171-224}$ activated P*vqmR-lux* to approximately WT levels, whereas only $VqmA_{Vc}{}^{*171-224}$ activated P*qtip-lux*, albeit weakly (Fig 5C and 5D, respectively). Consistent with this result, $VqmA_{Vc}{}^{*171-224}$ induced partial lysis in the *V. cholerae* phage VP882 lysogen (Fig 5E). $VqmA_{Vc}{}^{*171-224}$ was produced at levels similar to

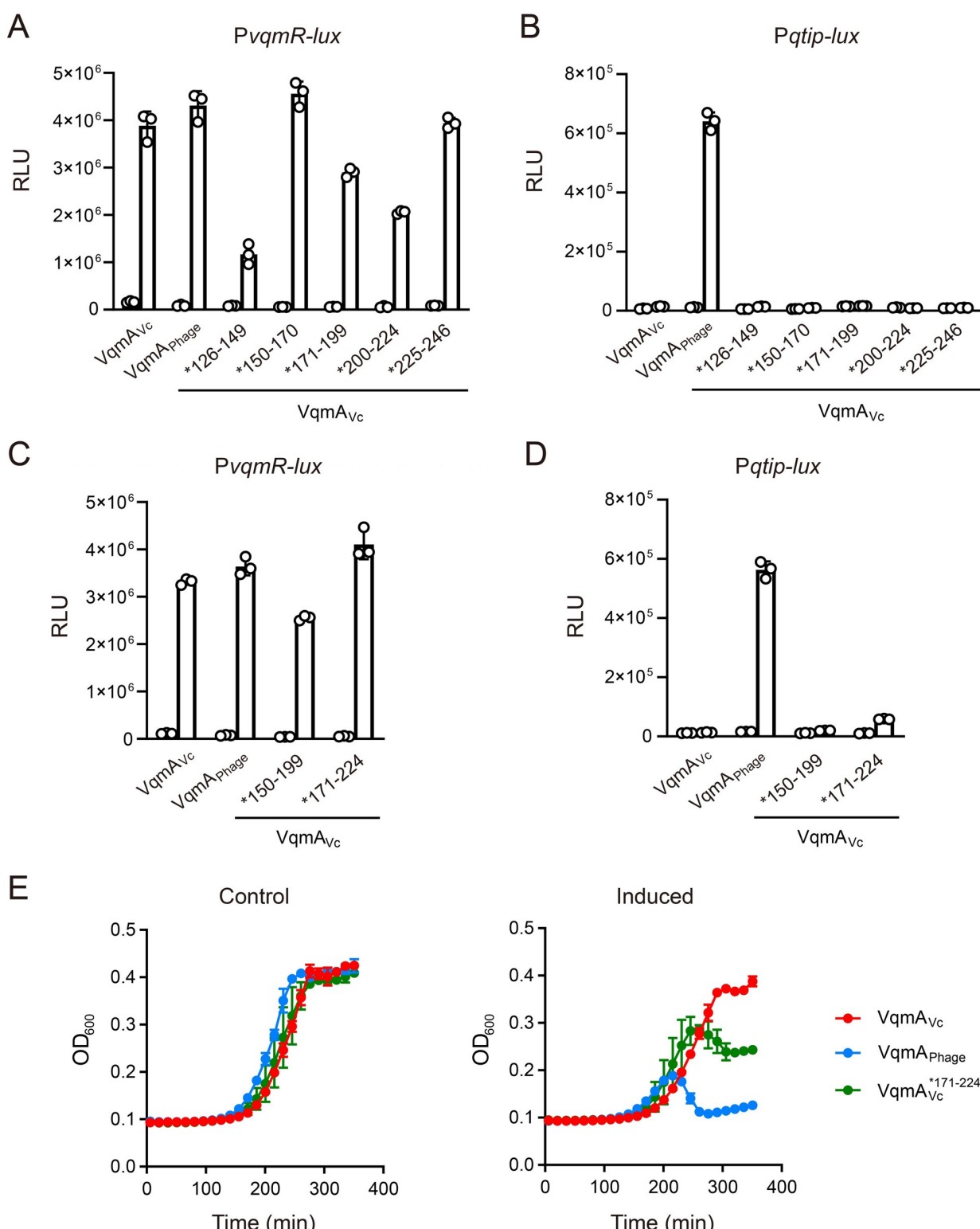

**Fig 5. The VqmA$_{Vc}$ HTH motif and the immediate C-terminal 25 residues, together, constrain binding to P*vqmR*.** (A-D) Normalized reporter activity from Δ*tdh E. coli* harboring (A and C) P*vqmR-lux* or (B and D) P*qtip-lux* and arabinose-inducible VqmA$_{Vc}$, VqmA$_{Phage}$, or the indicated VqmA$_{Vc}$ allele. Data are represented as mean ± SD (error bars) with $n = 3$ biological replicates. Black, no arabinose; white, 0.2% arabinose. (E) Growth curves of Δ*tdh* Δ*vqmA$_{Vc}$ V. cholerae* harboring phage VP882 *vqmA$_{Phage}$*::Tn5 and VqmA$_{Vc}$, VqmA$_{Phage}$, or VqmA$_{Vc}$*171–224 in medium lacking (Control) or containing 0.2% arabinose (Induced).

WT VqmA$_{Vc}$, eliminating the possibility that the observed binding to P*qtip* was a consequence of overexpression (S7 Fig). We conclude that the region encompassing both the HTH motif and the C-terminal 25 residues are required to restrict the VqmA$_{Vc}$ DBD from binding P*qtip*.

## Discussion

The DPO-VqmA QS AI-receptor pair controls lifestyle transitions in the pathogen *V. cholerae* and in the vibriophage VP882. Here, we studied the DNA-binding function of VqmA. VqmA proteins are cytoplasmic transcription factors composed of N-terminal PAS domains responsible for binding the DPO ligand and C-terminal DBDs containing HTH motifs. Most of the key residues required for binding the DPO ligand and for binding to P*vqmR* DNA are conserved between the two VqmA proteins. Indeed, both VqmA$_{Vc}$ and VqmA$_{Phage}$ bind DPO and activate transcription of *vqmR*. By contrast, only VqmA$_{Phage}$ activates the phage gene *qtip*. Here, we investigated this asymmetric DNA-binding pattern. Our work shows that, in both proteins, the DBD determines promoter recognition. We have previously shown that DPO binding enhances VqmA transcriptional activity [9]. This earlier work, together with our present results, suggest a model in which the PAS domain specifies DNA-binding affinity (between the apo- and holo- states), and the DBD specifies DNA-binding selectivity.

The main goal of the present work was to discover features of the VqmA proteins that confer specificity in transcriptional activity. We propose that phage VP882 VqmA$_{Phage}$ possesses a feature that relaxes its DNA-binding specificity and *V. cholerae* VqmA$_{Vc}$ possesses a feature that restricts its DNA-binding capability. Regarding VqmA$_{Vc}$, our genetic analyses support the hypothesis that the VqmA$_{Vc}$ DBD harbors elements that prevent it from binding P*qtip*. This hypothesis stems from our finding that residues G201, A202, E207, and M211 are crucial for VqmA$_{Phage}$ recognition of P*qtip*. These residues are conserved between VqmA$_{Vc}$ and VqmA$_{Phage}$. Specifically, in VqmA$_{Vc}$ they are: G199, A200, Q205, and L209, respectively. More broadly, sequence alignments of VqmA proteins among *Vibrios* reveal that the residue at the 207[th] position in VqmA$_{Phage}$ (205[th] position in VqmA$_{Vc}$) is most frequently either a Glu or a Gln [5]. Similarly, the residue at the 211[th] position in VqmA$_{Phage}$ (209[th] position in VqmA$_{Vc}$) is commonly a hydrophobic residue, like Met, Leu, Ile, or Val. Thus, E207 and M211 are not unique to VqmA$_{Phage}$, but rather occur in most VqmA proteins. We propose that because the key residues for P*qtip* binding are conserved in VqmA$_{Phage}$, VqmA$_{Vc}$, and other *Vibrio* VqmA proteins, VqmA$_{Vc}$ is likely restricted from binding P*qtip* by additional features elsewhere in its DBD. Regarding VqmA$_{Phage}$, the DPO-VqmA$_{Phage}$ structure was reported during review of this manuscript [15]. Superimposition of this new structure (7DWM) onto the DPO-VqmA$_{Vc}$ and DPO-VqmA$_{Vc}$-P*vqmR* structures (6KJU and 6IDE, respectively, and [9,11,14]) reveals two insights (S8 Fig). First, the conformations of the three PAS domains are similar except for the orientations of the first 20 N-terminal residues in each protein, indicating that the PAS domains do not confer the differences in promoter DNA specificity. Second, the DPO-VqmA$_{Phage}$ DBDs adopt a conformation that is intermediate between that of the more open DBDs in the DPO-VqmA$_{Vc}$ structure and the closed DBDs in the DPO-VqmA$_{Vc}$-P*vqmR* structure. Additionally, the interaction interface between the VqmA$_{Phage}$ DBDs is less extensive, and thus more relaxed than that of the VqmA$_{Vc}$ DBDs [15]. Likely, the more relaxed conformation exhibited by the VqmA$_{Phage}$ DBDs underpins its promiscuity for promoter binding with respect to P*vqmR* and P*qtip*.

In the case of VqmA$_{Phage}$, the residues G201, A202, E207, and M211 identified in our mutagenesis screen as necessary for P*qtip* binding are, surprisingly, not in the HTH motif, nor do the corresponding VqmA$_{Vc}$ residues make direct contacts with DNA in the DPO-VqmA$_{Vc}$-P*vqmR* crystal structure (Fig 4D). Thus, we wonder how the G201, A202, E207, and M211

residues could govern recognition of P*qtip*. Our *in vivo* analyses showed that substitutions in VqmA$_{Phage}$ at these residues enable activation of *vqmR* expression to WT levels, whereas only residual activation of *qtip* expression occurs (Fig 4A–4C). Surprisingly, the purified VqmA$_{Phage}$ mutant proteins maintained some capability to bind P*qtip in vitro*. A representative experiment using the VqmA$_{Phage}$$^{G201D}$ protein is shown in S9A Fig.

We consider several possibilities to explain our findings:

First, the VqmA$_{Phage}$ G201, A202, E207, and M211 residues could mediate interactions with an additional bacterial factor involved in transcription. Importantly, the failure of these VqmA$_{Phage}$ variants to activate P*qtip* expression in *V. cholerae* lysogens also occurred in *E. coli*, eliminating the possibility that these residues interact with a phage-specific or *Vibrio*-specific factor. Rather, these residues could be important for coordinating interactions with a conserved factor, such as RNA polymerase. If so, these mutant VqmA$_{Phage}$ proteins, while capable of binding promoter DNA, are incapable of activating transcription. This situation would be analogous to the positive control mutants of the lambda phage cI repressor (cI$_{lambda}$). So called pc mutants bind DNA and exhibit repressor activity, but are deficient in positive transcriptional regulation due to the inability of the mutant cI$_{lambda}$ proteins to productively interact with RNA polymerase [16,17]. In our case, the VqmA$_{Phage}$ mutants maintain the capacity to activate *vqmR* expression so they must successfully interact with RNA polymerase at least at P*vqmR*. For this reason, we consider it unlikely that these VqmA$_{Phage}$ mutants are analogous to lambda pc mutants.

Second, a global transcriptional regulator could be involved that is present in both *V. cholerae* and *E. coli*. One candidate is the histone-like nucleoid structuring protein (H-NS) that functions as a universal repressor of transcription [18]. In *Vibrio harveyi*, the QS master regulator, LuxR, displaces H-NS at promoter DNA to activate expression of QS-controlled genes [19]. Perhaps, the VqmA$_{Phage}$ G201, A202, E207, and M211 mutants cannot successfully compete with H-NS for binding at P*qtip in vivo*, whereas in an EMSA assay, since H-NS is not present, binding to P*qtip* DNA occurs. To address this possibility, we examined whether WT VqmA$_{Phage}$ and VqmA$_{Phage}$$^{G201D}$ competed with H-NS for binding to P*qtip* using EMSA assays. There was no difference between WT VqmA$_{Phage}$ and VqmA$_{Phage}$$^{G201D}$ binding to P*qtip* DNA in the presence of purified H-NS (S9C and S9D Fig). These experiments suggest that it is unlikely that H-NS competition underlies our findings.

Third, the binding of the VqmA$_{Phage}$ G201, A202, E207, and M211 mutants to P*qtip in vitro*, while demonstrating loss of activity *in vivo*, could be a consequence of the unnaturally high DNA: VqmA$_{Phage}$ stoichiometry in the EMSA, similar to what we observed for the GST-DBD$_{Vc}$ construct (Fig 1G). Thus, the EMSA is not sufficiently sensitive to distinguish between the strength of DNA binding of WT VqmA$_{Phage}$ and the residual binding by the VqmA$_{Phage}$ G201, A202, E207, and M211 mutants. If this is the case, we propose that VqmA$_{Phage}$ G201, A202, E207, and M211 could play allosteric roles in correctly positioning the VqmA$_{Phage}$ HTH for proper contact with particular DNA nucleotides. Here, we compare this possibility to how site-specific recognition is accomplished by cI$_{lambda}$. Genetic and biochemical studies revealed that residues outside of the cI$_{lambda}$ HTH motif are crucial for site-specific DNA recognition [20–24]. The crystal structure of the cI$_{lambda}$ repressor bound to DNA shows that charged residues adjacent to those in the HTH interact with the DNA sugar phosphate backbone [25]. Additionally, the N-terminal arm of cI$_{lambda}$ wraps around the DNA and makes contacts on the backside of the helix [25]. It is presumed that the backbone contacts function to position the HTH residues to contact specific DNA nucleotides. Thus, while the VqmA$_{Phage}$ residues that we identified as important for P*qtip* recognition (G201, A202, E207, and M211) do not function perfectly analogously to those in cI$_{lambda}$ because they do not make contact with the DNA backbone, their role in site-specific recognition could be similar. A

caveat of our interpretation is that, as noted, we do not have a structure of VqmA$_{Phage}$ bound to P*qtip* and we mapped the residues identified in our VqmA$_{Phage}$ mutagenesis to the DPO-VqmA$_{Vc}$-P*vqmR* crystal structure. Therefore, it remains possible that the residues we identified here do indeed make contacts with DNA. A further possibility is that the residues we identified foster increased plasticity to the VqmA$_{Phage}$ DBDs, perhaps, allowing VqmA$_{Phage}$ to bind the longer palindrome that exists in P*qtip*, which we discuss below. The recently reported DPO-VqmA$_{Phage}$ crystal structure [15], together with the existing DPO-VqmA$_{Vc}$ structures, could enable modeling to predict the roles played by particular residues in conferring a relaxed conformation to the VqmA$_{Phage}$ DBDs. To our knowledge, no region analogous to the one we discovered in VqmA$_{Phage}$ has been shown to confer promoter specificity to a transcription factor. Going forward, determining the structure of VqmA$_{Phage}$ bound to P*qtip* DNA should reveal the mechanism enabling recognition of P*qtip* and the role that these residues play, individually and collectively, in determining DNA-binding specificity.

Previous work demonstrated that VqmA$_{Vc}$ recognizes a key GG-N$_6$-CC palindrome in P*vqmR* [2,11]. Our sequence alignment of P*vqmR* and P*qtip* showed that P*qtip* does not possess this palindrome. Rather, the corresponding sequence in P*qtip* is GG-N$_6$-TA (Fig 2A). The most obvious divergence between the two sequences is in the central six nucleotides: "-AAAATA-" in P*qtip* and "-TTTCCC-" in P*vqmR* (Fig 2A). We hypothesized that this nucleotide stretch could be responsible for conferring the asymmetric DNA-binding patterns to the two VqmA proteins. Indeed, exchanging these nucleotides in P*qtip* and P*vqmR* reversed the promoter binding preferences of the VqmA proteins. We verified our conclusion that this core 6 nucleotide stretch drives VqmA DNA-binding preference using our VqmA chimeric proteins ($_{Vc}$N-C$_{Phage}$ and $_{Phage}$N-C$_{Vc}$), a representative mosaic protein (VqmA$_{Vc}$*$^{171-224}$), and a representative protein containing a point mutation (VqmA$_{Phage}$$^{G201D}$) (S9A, S9B, and S10 Figs). While the present manuscript was under review, Gu *et al.* reported that a GG-N$_9$-CC palindrome in P*qtip* is the key sequence for VqmA$_{Phage}$ recognition [15]. According to our DNA sequence alignment, the GG-N$_6$-CC palindrome required for VqmA$_{Vc}$ binding is only present in P*vqmR*, while the key GG-N$_9$-CC palindrome required for VqmA$_{Phage}$ binding exists in both P*qtip* and P*vqmR* (Fig 2A). Together, our results and those of Gu *et. al.* [15] explain, at the level of the promoter DNA, why VqmA$_{Phage}$ binds both P*qtip* and P*vqmR* while VqmA$_{Vc}$ recognizes only P*vqmR*.

Genomic sequencing data have revealed the presence of many QS receptor-transcription factors encoded in phage genomes [26]. In general, however, their transcriptional outputs are uncharacterized, with the exception of VqmA$_{Phage}$, which is promiscuous with respect to binding to P*vqmR* and P*qtip*, the only two promoters tested to our knowledge. It remains possible that VqmA$_{Phage}$ regulates additional genes specifying bacterial and or/phage functions. Given that VqmA$_{Phage}$ can regulate biofilm formation through its control of *V. cholerae vqmR*, probing the host regulon controlled by VqmA$_{Phage}$ under various growth conditions could reveal unanticipated roles of QS in phage-*Vibrio* interactions.

Finally, we found that the VqmA$_{Vc}$$^{A192E}$ variant exhibited modest, but detectable binding to P*qtip*, whereas the VqmA$_{Vc}$ quadruple mutant, and the VqmA$_{Vc}$*$^{171-199}$ mosaic protein did not. Western blot and P*vqmR-lux* assays eliminated the possibility that any of the mutant proteins were not expressed or were misfolded. Rather, we infer that a particular regional conformation in the VqmA proteins is required for this key residue to function properly. Our results also show that exchanging both the VqmA$_{Vc}$ HTH motif and C-terminal 25 residues with the corresponding residues from VqmA$_{Phage}$ enables some but not WT-level binding to P*qtip*. This finding supports the notion that a set of non-contiguous amino acids or a particular conformation of the VqmA$_{Vc}$ DBD prevents binding to P*qtip*. This arrangement is perhaps not surprising given that *V. cholerae* would pay a significant penalty if VqmA$_{Vc}$ bound the phage

VP882 *qtip* promoter, as the consequence would be the launch of the phage lytic program and death of the host cell. To our knowledge, VqmA$_{Vc}$ binds to only one promoter, P*vqmR* [3]. Thus, even in the context of the *V. cholerae* genome, VqmA$_{Vc}$ transcriptional activity is tightly constrained. It is possible that other negative ramifications stem from non-specific VqmA$_{Vc}$ binding in the *V. cholerae* genome. Distinct mechanisms are employed to restrict other QS receptor/transcription factors from promiscuously binding to DNA. For example, LuxR-type QS receptors can typically bind >100 promoters, but their solubilization, stability, and DNA-binding capabilities strictly rely on being bound to an AI whose availability is, in turn, highly regulated [27–31]. Therefore, precise control of gene expression is maintained in many QS circuits by confining QS receptor activity to the ligand-bound form coupled with discrete affinities of the ligand-receptor complexes for target promoters. By contrast, VqmA$_{Vc}$ is expressed constitutively, and its DNA-binding capabilities are not limited by the presence of an AI. Thus, exquisitely tight control over promoter DNA-binding specificity by VqmA$_{Vc}$—restricting it to one and only one promoter—is apparently crucial for proper regulation of gene expression and survival.

## Materials and methods

### Bacterial strains, plasmids, primers, and reagents

Strains, plasmids, primers, and gBlocks used in this study are listed in S1–S4 Tables, respectively. In all experiments, Δ*tdh V. cholerae* and Δ*tdh E. coli* strains were used except in the experiment assaying expression of P*vqmR-lux* and P*qtip-lux* in response to the DBD$_{Vc}$, DBD$_{Phage}$, GST-DBD$_{Vc,}$ and GST-DBD$_{Phage}$ proteins. In that case, the *E. coli* strain contained the WT *tdh* gene. *V. cholerae* and *E. coli* were grown aerobically in lysogeny broth (LB) at 37°C. Antibiotics and inducers were used at the following concentrations: 50 units mL$^{-1}$ polymyxin B, 200 μg mL$^{-1}$ ampicillin, 5 μg mL$^{-1}$ chloramphenicol, 100 μg mL$^{-1}$ kanamycin, 0.2% arabinose, and 1 mM Isopropyl β-D-1-thiogalactopyranoside (IPTG).

Primers were obtained from Integrated DNA Technologies. Gibson assembly, intramolecular reclosure, and traditional cloning methods were employed for all cloning. PCR with Q5 High Fidelity Polymerase (NEB) was used to generate insert and backbone DNA. Gibson assembly relied on HiFi DNA assembly mix (NEB). All enzymes used in cloning were obtained from NEB. Mutageneses of the VqmA$_{Phage}$ and VqmA$_{Vc}$ DBDs were accomplished using the GeneMorph II EZClone Domain Mutagenesis Kit (Agilent) according to the manufacturer's instructions. Transfer of plasmids carrying *vqmA* genes into the *V. cholerae* phage VP882 lysogen employed conjugation followed by selective plating on polymyxin B, chloramphenicol, and kanamycin, based on previously described protocols [32].

### Genetic screens for VqmA$_{Phage}$ and VqmA$_{Vc}$ DNA-binding mutants

*E. coli* carrying a library of plasmid-borne *vqmA$_{Phage}$* mutants was mated with *V. cholerae* harboring a phage VP882 mutant (*vqmA$_{Phage}$*::Tn5) and the P*vqmR-lux* reporter integrated at the *lacZ* locus. Exconjugant *V. cholerae* colonies were collected and streaked onto LB agar plates supplemented with polymyxin B, chloramphenicol, kanamycin, and arabinose. P*vqmR-lux* activity of surviving exconjugants was assayed using an ImageQuant LAS4000 imager (GE). *V. cholerae* colonies that produced light were harvested for plasmid DNA preparation. Isolated plasmid DNA was subsequently transformed into *E. coli* strains carrying P*qtip-lux* or P*vqmR-lux* to validate activity.

A library of plasmid-borne *vqmA$_{Vc}$* mutants was transformed into *E. coli* carrying the P*qtip-lux* reporter. Transformants were plated on LB agar supplemented with ampicillin,

kanamycin, and arabinose. P*qtip-lux* activity was assayed using an ImageQuant LAS4000 imager.

## Growth, lysis, and bioluminescence assays

To measure growth of *V. cholerae* phage VP882 lysogens or activation of the P*vqmR-lux* and P*qtip-lux* reporters in bacterial strains, overnight cultures of *V. cholerae* or *E. coli* were back-diluted 1:1000 into LB medium supplemented with appropriate antibiotics prior to being dispensed (200 μL) into 96-well plates (Corning Costar 3904). Arabinose was added as specified. The plates were shaken at 37˚C and a Biotek Synergy Neo2 Multi-Mode reader was used to measure $OD_{600}$ and bioluminescence. For bioluminescence assays, relative light units (RLU) were calculated by dividing bioluminescence by the $OD_{600}$ after 5 h.

## Protein expression, purification, and electrophoretic mobility shift assay (EMSA)

Protein expression and purification were performed as described [9,19]. EMSAs were performed as described [8] with the following modifications: Following electrophoresis, 6% DNA retardation gels were stained with SYBR Green (Thermo) and visualized using an ImageQuant LAS 4000 imager with the SYBR Green settings. Unless specified otherwise, the highest concentration of VqmA assessed was 600 nM. 25 nM P*vqmR* or P*qtip* DNA was used in all EMSAs. The percentage of promoter DNA bound was calculated using the gel analyzer tool in ImageJ and the estimated $EC_{50}$ values were derived from $EC_{50}$ analyses in Prism.

## Western blot analysis

Western blot analyses probing for abundances of 3xFLAG-tagged proteins were performed as reported [3] with the following modifications: *E. coli* and *V. cholerae* carrying N-terminal 3xFLAG-tagged VqmA$_{Vc}$ and N-terminal 3xFLAG-tagged VqmA$_{Phage}$ alleles were back-diluted 1:1000 in LB supplemented with appropriate antibiotics and harvested after 6 h and 4 h of growth at 37˚C, respectively. Cells were resuspended in Laemmli sample buffer at a final concentration of 0.006 OD/μL. Following denaturation for 15 min at 95˚C, 5 μL of each sample was subjected to SDS-PAGE gel electrophoresis. RpoA was used as the loading control (Biolegend Inc.). Signals were visualized using an ImageQuant LAS 4000 imager.

## Sequence alignments

Protein and DNA sequences in FASTA format were aligned in the BioEdit Sequence Alignment Editor using the default setting under the ClustalW mode. Figs 2A and S1 were prepared via the ESPript 3.0 online server [33].

## Statistical methods

All statistical analyses were performed using GraphPad Prism software. Error bars correspond to standard deviations of the means of three biological replicates.

## Supporting information

**S1 Fig. Sequence alignment of VqmA proteins.** Protein sequence alignment (ClustalW) showing VqmA$_{Vc}$ and VqmA$_{Phage}$. Black and white boxes designate identical and conserved residues, respectively. The PAS domain and HTH motif are indicated. The site used to fuse domains for chimera constructions is indicated by the red box. Key residues required for DPO binding are designated with black triangles. Conserved HTH residues are designated by black

circles and open circles show residues with different charges in the HTH motifs of the two proteins. The residue in each HTH motif that contributes to P*qtip* specificity is designated by the striped circle. The residues identified in the VqmA$_{Phage}$ screen and the equivalent residues altered by site-directed mutagenesis in VqmA$_{Vc}$ are designated by asterisks.
(TIF)

**S2 Fig. VqmA$_{Phage}$ has higher affinity for P*qtip* DNA than for P*vqmR* DNA.** (A) EC$_{50}$ analysis of the designated VqmA proteins for binding to P*vqmR* and P*qtip*. Data are representative of two independent experiments. The percentage of DNA bound was calculated using the gel analyzer tool in ImageJ and the estimated EC$_{50}$ values were derived from Prism. (B) Competitive VqmA$_{Phage}$ and $_{Vc}$N-C$_{Phage}$ EMSA analysis. 25 nM P*vqmR* and P*qtip* DNA were used and no protein (designated -) or 2-fold serially-diluted protein was added to the lanes. The lowest and highest protein (dimer) concentrations are 4.7 nM and 1200 nM, respectively.
(TIF)

**S3 Fig. The VqmA$_{Vc}$ and VqmA$_{Phage}$ DBDs are non-functional.** (A and B) Normalized reporter activity from WT *E. coli* harboring (A) P*vqmR-lux* or (B) P*qtip-lux* and arabinose-inducible VqmA$_{Vc}$, DBD$_{Vc}$, VqmA$_{Phage}$, and DBD$_{Phage}$. Black, no arabinose; white, 0.2% arabinose. Data are represented as mean ± SD (error bars) with *n = 3* biological replicates. (C) EMSAs of DBD$_{Vc}$ and DBD$_{Phage}$ proteins binding to P*vqmR* and P*qtip*. 25 nM P*vqmR* or P*qtip* DNA was used in all EMSAs with no protein (designated -) or 2-fold serial dilutions of proteins. The lowest and highest protein (dimer) concentrations are 18.75 nM and 600 nM, respectively. (D) Gel filtration chromatogram showing UV$_{280}$ traces for the purification of (left) VqmA$_{Vc}$, DBD$_{Vc}$, and GST-DBD$_{Vc}$ and (right) VqmA$_{Phage}$, DBD$_{Phage}$, and GST-DBD$_{Phage}$ proteins. (E) EMSA of GST protein binding to P*vqmR* and P*qtip* DNA as in panel C.
(TIF)

**S4 Fig. The VqmA$_{Phage}$ and VqmA$_{Vc}$ variants are produced at levels similar to WT.** Western blot showing the designated (A and C) 3xFLAG-VqmA$_{Phage}$ and (B) 3xFLAG-VqmA$_{Vc}$ proteins produced by Δ*tdh E. coli* and Δ*tdh* Δ*vqmA$_{Vc}$ V. cholerae*. A contaminating band below VqmA$_{Phage}$ and VqmA$_{Vc}$ is present in all Δ*tdh E. coli* samples. The RNAPα subunit (RpoA) was used as the loading control. Data are representative of two independent experiments.
(TIF)

**S5 Fig. VqmA$_{Phage}$ mutants possessing WT activity induce phage lysis on agar plates supplemented with 0.2% arabinose.** Shown is growth of Δ*tdh* Δ*vqmA$_{Vc}$ V. cholerae* harboring phage VP882 *vqmA$_{Phage}$*::Tn*5* as a lysogen and arabinose-inducible 3xFLAG-VqmA$_{Phage}$ streaked onto agar plates with no arabinose (Control) or 0.2% arabinose (Induced).
(TIF)

**S6 Fig. VqmA$_{Vc}$$^{Q205E}$ and VqmA$_{Vc}$$^{L209M}$ do not bind P*qtip*.** (A and B) Normalized reporter activity from Δ*tdh E. coli* harboring (A) P*vqmR-lux* or (B) P*qtip-lux* and arabinose-inducible 3xFLAG-VqmA$_{Vc}$, 3xFLAG-VqmA$_{Phage}$, or the indicated 3xFLAG-VqmA$_{Vc}$ allele. Black, no arabinose; white, 0.2% arabinose. Data are represented as mean ± SD (error bars) with *n = 3* biological replicates.
(TIF)

**S7 Fig. VqmA$_{Vc}$ mosaic proteins are produced at levels similar to WT VqmA$_{Vc}$.** Western blot showing the designated 3xFLAG-VqmA$_{Vc}$ mosaic proteins produced by Δ*tdh E. coli* and Δ*tdh* Δ*vqmA$_{Vc}$ V. cholerae*. RpoA was used as the loading control. Data are representative of

two independent experiments.
(TIF)

**S8 Fig. Structural comparisons of the VqmA$_{Phage}$ and VqmA$_{Vc}$ proteins.** Previously reported crystal structures of DPO-VqmA$_{Vc}$-P*vqmR* (blue, PDB: 6IDE) and DPO-VqmA$_{Vc}$ (green, PDB: 6KJU) superimposed onto the recently published crystal structure of DPO-VqmA$_{Phage}$ (yellow, PDB: 7DWM) based on the orientations of the PAS domains. DNA in the DPO-VqmA$_{Vc}$-P*vqmR* structure was omitted for simplicity.
(TIF)

**S9 Fig. EMSA analyses of the VqmA$_{Phage}$$^{G201D}$ protein binding to DNA.** (A) EMSA showing binding of VqmA$_{Phage}$$^{G201D}$ to P*vqmR* and P*qtip* DNA. 25 nM DNA was used in all EMSAs with no protein (designated -) or 2-fold serial dilutions of proteins. The lowest and highest protein (dimer) concentrations are 18.75 nM and 600 nM, respectively. (B) As in panel A for P*vqmR*\* and P*qtip*\* DNA. (C) EMSA showing WT VqmA$_{Phage}$ and VqmA$_{Phage}$$^{G201D}$ binding to P*qtip* DNA in the presence of H-NS (300 nM). (D) EMSA showing H-NS binding to P*qtip* DNA in the presence of WT VqmA$_{Phage}$ or VqmA$_{Phage}$$^{G201D}$ (each protein at 300 nM).
(TIF)

**S10 Fig. EMSA analyses of mosaic and chimeric VqmA proteins binding to P*vqmR*\* and P*qtip*\* DNA.** (A) EMSA showing binding of VqmA$_{Vc}$\*$^{171-224}$ to P*vqmR* and P*qtip* DNA. 25 nM DNA was used in all EMSAs with no protein (designated -) or 2-fold serial dilutions of proteins. The lowest and highest protein (dimer) concentrations are 18.75 nM and 600 nM, respectively. (B) As in panel A for P*vqmR*\* and P*qtip*\* DNA. (C) As in panel A for $_{Vc}$N-C$_{Phage}$ binding to P*vqmR*\* and P*qtip*\* DNA. (D) As in panel C for $_{Phage}$N-C$_{Vc}$.
(TIF)

**S1 Table. Bacterial strains used in this study.**
(DOCX)

**S2 Table. Plasmids used in this study.**
(DOCX)

**S3 Table. Primers used in this study.**
(DOCX)

**S4 Table. gBlocks used in this study.**
(DOCX)

**S1 Data. Numerical data for Figs 1A, 1B, 1C, 1E, 1F, 2C, 2D, 3A, 3B, 3C, 3D, 3E, 3F, 4A, 4B, 4C, 5A, 5B, 5C, 5D, 5E, S2A, S3A, S3B, S3D, S6A and S6B.**
(XLSX)

## Acknowledgments

We thank members of the Bassler laboratory for insightful discussions.

## Author Contributions

**Conceptualization:** Olivia P. Duddy, Xiuliang Huang, Justin E. Silpe, Bonnie L. Bassler.

**Data curation:** Olivia P. Duddy, Xiuliang Huang, Justin E. Silpe.

**Formal analysis:** Olivia P. Duddy, Xiuliang Huang, Justin E. Silpe, Bonnie L. Bassler.

**Funding acquisition:** Bonnie L. Bassler.

**Investigation:** Olivia P. Duddy, Xiuliang Huang, Justin E. Silpe, Bonnie L. Bassler.

**Methodology:** Olivia P. Duddy, Xiuliang Huang, Justin E. Silpe.

**Project administration:** Bonnie L. Bassler.

**Resources:** Olivia P. Duddy, Xiuliang Huang, Justin E. Silpe.

**Supervision:** Bonnie L. Bassler.

**Validation:** Olivia P. Duddy, Xiuliang Huang, Justin E. Silpe, Bonnie L. Bassler.

**Visualization:** Olivia P. Duddy, Xiuliang Huang, Justin E. Silpe.

**Writing – original draft:** Olivia P. Duddy, Xiuliang Huang, Justin E. Silpe, Bonnie L. Bassler.

**Writing – review & editing:** Olivia P. Duddy, Xiuliang Huang, Justin E. Silpe, Bonnie L. Bassler.

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
