## [Decision Letter · Decision Letter 0]

10 May 2021

Dear Dr Bassler,

Thank you very much for submitting your Research Article entitled 'Mechanism underlying the DNA-binding preferences of the Vibrio cholerae and vibriophage VP882 VqmA quorum-sensing receptors' to PLOS Genetics.

The manuscript was fully evaluated at the editorial level and by three independent peer reviewers. The reviewers appreciated the attention to an important and interesting problem. They agreed that the presented experimental data are high quality and that the manuscript is well written, but had differing opinions regarding the suitability of the manuscript for publication in PLoS Genetics. Based on the reviews, we will not be able to accept this version of the manuscript, but we would be willing to review a revised version.

Should you choose to revise the manuscript for further consideration here, you should provide a detailed list of your responses to the review comments and a description of the changes you have made in the manuscript. Revisions that address reviewers’ concerns regarding quantitation and interpretation of VqmA DNA binding data, and that address reviewers’ questions regarding how the central DNA binding site region determines specificity should be seriously considered. The very recently published structure 7DWM by Gu et al will likely be useful as you revise.

If you decide to revise the manuscript for further consideration at PLOS Genetics, please aim to resubmit within the next 60 days, unless it will take extra time to address the concerns of the reviewers, in which case we would appreciate an expected resubmission date by email to plosgenetics@plos.org.

[LINK]

We are sorry that we cannot be more positive about your manuscript at this stage. Please do not hesitate to contact us if you have any concerns or questions.

Yours sincerely,

Sean Crosson

Associate Editor

PLOS Genetics

Lotte Søgaard-Andersen

Section Editor: Prokaryotic Genetics

PLOS Genetics

Reviewer's Responses to Questions

**Comments to the Authors:**

Reviewer #1: Duddy et al present a detailed mechanistic evaluation of the differences in function between two VqmA transcription factors: one from Vibrio cholerae that binds DPO and one encoded by phage VP882 that also binds DPO. This manuscript explores the amino acid differences between the VqmA proteins and the nucleic acid differences between the Pqtip and PvqmR binding sites that result in VqmAphage binding to both substrates but restrict VqmAVc to only binding PvqmR. Their conclusions are: 1) the 6 central nucleotides in the DNA binding sites drive promoter specificity, 2) VqmAphage and VqmAVc have specific DNA binding domains, whereas their PAS and dimerization domains are interchangeable, 3) two regions of VqmA are required to interact specifically with Pqtip.

This was a really interesting read on a relevant topic. The experimental design was very clear and impressive in dissecting the mechanism of VqmA DNA binding that differs between the bacterium and phage. Overall, this was a clearly and expertly written paper. There are some wording choices that are not always substantiated with direct evidence (i.e., lack of quantified EMSAs, statistical analyses, etc.).

In addition, the authors should be aware that another publication recently came out that overlaps with some of the experiments in this manuscript:

Understanding the mechanism of asymmetric gene regulation determined by the VqmA of vibriophage.

Gu Y, Zhi SX, Yang N, Yang WS.

Biochem Biophys Res Commun. 2021 Apr 22;558:51-56. doi: 10.1016/j.bbrc.2021.04.036. Online ahead of print.

PMID: 33895551

Experiments that overlap:

• Chimeras with PAS domain swaps

• Modulation of Pqtip DNA binding site

Gu et al. also report a crystal structure of VqmAPhage-DPO. This will aid the authors in mapping the substituted VqmAphage residues from the random mutagenesis screen.

Major comments:

• Line 91 and 286 – “VqmAVC must possess an additional feature that prevents it from binding Pqtip”; couldn’t VqmAPhage possess the additional feature and VqmAVC be lacking said feature? The authors repeatedly say this isn’t the case, but how have the authors excluded that possibility? Which data specifically show that VqmAphage doesn’t have the additional feature that enables recognition of the 6 core nucleotides? Given the findings in the new study by Gu et al., the X-ray crystal structure indicates a more relaxed conformation on binding with DPO. Perhaps this is the feature that enables recognition?

• Line 138-139 - “the respective DBD of each purified VqmA protein drives promoter selectivity.”; The C-terminal swap does not appear to be the only driver of promoter selectivity here as can be seen with the “~7-fold lower affinity”. Is the 7-fold lower affinity referring to the lux results? First, I think quantifying these shifts (in triplicate) would really help here with comparing binding of these chimeras. Second, is some sort of additional allostery at play here (besides the DPO ligand-dependent dimerization)? The dimerization experiments do indeed show that promoter-binding selectivity is mostly mediated by the DBD, but I do not believe it is “governed exclusively by the DBD” as mentioned on Line 168. This claim seems too strong, especially given the data shown in later figures. The decreases in DNA binding when using the GST dimerization method shows it cannot just be the DBD, at least not in the context shown here. I suggest that the authors either weaken wording or substantiate their claim with quantified EMSA binding data.

• Line 186 – “VqmAVC only shifted Pqtip*”; this protein still appears to have a slight shift of PvqmR at the two highest concentrations (300 and 600 nM I believe). There is a distinct band in the bound region for the 300 nM condition. Change wording or consider repeating EMSAs with BSA or other protein to decrease non-specific binding, if that is indeed what is happening in that gel shift.

• Line 192-193 – consider replacing “is sufficient to prevent VqmAVC from binding” to a more accurate statement based on the EMSAs and transcriptional fusion assays, such as “decreases VqmAVC binding affinity” or “prevents VqmAVC transcription at Pqtip.”

• Line 213-214 – It does not appear that all four VqmAPhage single point mutants possessed WT activity (unless after statistical analysis it can be shown that there is no significant difference between these activities.)

• Figures 3B, 3C, 3E, and 3F would benefit from statistical analysis to help solidify some of the claims made in the text.

• Lines 282-287: The section discussing the conclusions of the random mutagenesis of the VqmAphage DBD is reaching. Point 2 stating that the four residues in VqmAphage must not play any role cannot be stated from these data. Only a couple substitutions were tested, and one could argue that the lack of a bulky or charged amino acid in place of the existing amino acids is “playing a role”. Point 3: why do the authors state that VqmAVc has the capacity to bind Pqtip? I understand that analogous amino acids exist in VqmAphage, but at this point there are no defined data showing that it has the ability to bind Pqtip. This statement is odd, given that the protein itself doesn’t bind Pqtip, which the authors have shown experimentally.

• It would help for the authors to discuss more details about the failed mutagenesis in which VqmAVc mutants were not able to activate Pqtip. How many mutants did they screen? What was the average mutagenesis rate per allele? Of course, it is possible that by random mutagenesis the number of substitutions required to enable VqmAVc to bind Pqtip may never work. But this is hard to assess without more details.

• The authors nicely show that the 6 core nucleotides drive specificity – great experiment! But the authors don’t comment on the residues shown to contact these bases in the DNA-bound VqmAVc structure: Gln174, Lys172, Lys203, Glu188, Arg195, Lys185, Tyr186. The VqmAVc171-199 mosaic should have these all. Could the authors comment on this inconsistency in these two results?

Minor comments:

• Line 33: “as” necessary sounded awkward…perhaps “that is”

• Line 50: if VqmAvc is overexpressed…will it bind the phage BS?

• Line 273: specify that it is PvqmR DNA in the structure

Reviewer #2: Bassler and coworkers have previously investigated the behavior of a prophage-encoded gene, VqmAphage, a homolog of a chromosomally encoded gene (VqmA) in Vibrio cholerae. Both genes are transcriptional activators that become active in the presence of the quorum-sensing signal DPO; the phage gene turns on an anti-repressor, leading to phage induction and lysis of its host. Here, the authors investigate the basis for asymmetric binding and activity of these two proteins; while both activate the host target gene vqmR, only the phage gene can act on the phage target (qtip). Via chimeric targets and proteins, the authors demonstrate specificity determinants in both target and protein, and also find that the bacterial VqmA contains multiple sequences that interfere with its ability to activate the prophage promoter. The work provides a range of insights into specificity determinants, although, as noted, actually understanding how these determinants work will likely require structures.

1. In terms of the logic of the regulatory circuit, it seems reasonable that the host gene cannot activate phage lysis. When, however, would it be particularly useful for the phage, as it starts into a lytic cycle, to activate the host vqmR gene?

2. Line 79-81: This idea that it is surprising a regulator “can recognize …promoters… present in genes in different kingdoms” seems like over-selling this story. After all, while one can say the virus is in a different kingdom, it is entirely dependent on the bacteria for growth, and phage (and animal viruses) clearly manipulate host machinery in many ways.

3. Introduction, lines 82-98: This reiteration of everything in the paper is much more detailed then is needed here, and as a result doesn’t emphasize the most important points. Why list all the specific residues here?

4. Lines 110-117: Please clarify why these experiments were all done in the absence of DPO. I did not understand the explanation offered here. DPO is not needed for DNA binding? This is implied, but not said. Is the activation being measured in this paper different in other requirements to what is seen with DPO? Or does that simplify amplify signals, and overproducing the proteins under arabinose induction leads to the same response? In particular, given the focus here, is the mode of activation the same?

5. P. 7, lines 140-147: The question about the PAS domain playing a role in promoter binding selectivity may be reasonable, but given that removing it renders the proteins entirely inactive, the way this is set up is not useful. I would suggest starting with the question posed more generally, or just a simple statement that deleting the PAS domain led to inactive proteins. Then the suggestion that this might be because of a requirement for dimerization can be made and tested. I’m not convinced by the GST results that one can rule out a role for PAS, given the authors’ finding later that sequences elsewhere in the protein have a restrictive effect on DNA binding patterns.

6. The demonstration that a change of a central region in the VqmA binding region form AAAATA to TTTCCC is striking and could be further investigated. For instance, this result might suggest that the angle/bending between the dimer binding sites for VqmA plays an important role in the specificity of binding of the different proteins. Was this looked at all? Would changing spacing even slightly change binding specificity? It would be useful to include in Figure 2 where the start point for transcription is, where -10, -35 sequences are. If this is not relevant (see query 4 above), please provide a clear explanation of why not.

7. Do the various chimeras and mutants all show the same discrimination based on the central region of the binding site? Is this the only specificity determinant in the promoter? Mutant proteins might help to determine that; certainly in Fig. 2C, D, changing the site doesn’t totally lead to VqmAVC acting like the phage protein on pqtip, just relieves the stringency modestly.

8. Fig. 3B, C, Line 215: The authors focus here on E194A as the least active on the qtip promoter but R164I looks equally defective in the promoter activity assay. The basis for concluding (as in line 220-221) that only E194 matters is therefore only due to its less extreme lysis effect? The fact that these don’t fully correlate might suggest an addition role for VqmAphage in the lysis pathway; has that been ruled out?

9. Line 248: clarify that the mutation library is in the context of the otherwise full-length protein.

10. Line 282-287: The conclusions reached here need more explanation and could be rewritten. Was the mutagenesis and selection/screening method for finding these four residues saturating? The first two conclusions follow from the way these were identified, requiring that they still activate vpmR; maybe make that more explicit here? Where does the conclusion that both proteins “possess the capacity to bind pqtip DNA” come from? That these residues are identical in VqmA? Why is this evidence of the capacity to bind? Maybe less definitive conclusions and a suggestion that because mutations which interfere with qtip binding by VqmAphage are not in residues only found in the phage protein, likely specificity lies elsewhere. Then the suggestion that this is in the restriction of VqmA binding, rather than (or in addition to) positive elements in VqmAphage.

11. I’m assuming that this 171-224 construct still has the rest of the sequence past 224 to the end of VqmA. Is that correct? Was a simple truncation of the C-terminal amino acids, maybe ending with the K found in VqmAphage, ever tested? How does the +171-224 VqmA construct act on the slightly permissive DNA binding site chimera (pqtip*)?

12. Discussion: Wouldn’t it be fairly easy to see if removing hns leads to more permissive VqmA(phage mutants or host) binding/activity?

13. Discussion: Is there any cost to the host of expressing the VqmAphage in cells not carrying a prophage? Would that help to address whether there is any other promiscuous activity of this protein?

14. Figure 1 D, G, elsewhere: I found it somewhat confusing to have the triangles indicating increasing amounts of VqmA protein labeled with the DNA it is binding to. Maybe put the pvqmR, etc. labels below the gel, just have the relevant protein above?

Reviewer #3: In the study " Mechanism underlying the DNA-binding preferences of the Vibrio cholerae and vibriophage VP882 VqmA quorum-sensing receptors", the Dr Bassler’s group use a mutagenic approach to pinpoint the molecular determinants of VqmA transcriptional factor selectivity. VqmA transcriptional factor is a quorum-sensor receptor of DPO autoinducer that in Vibrio cholerae controls genes involved in biofilm formation and virulence factor production. Recently, this same group described that the phage VP882, which infect V. cholerae, also encodes for a VqmA receptor (VqmA-phage) that is similarly regulated by DPO. VqmA-phage controls the transcription of the phage gene qtip, which regulates lytic program, but it is also able to regulate the same genes that the bacterial host VqmA (VqmA-Vc)i regulates. In contrast, VqmA-Vc is not able to regulate qtip, thus showing a transcriptional asymmetry between both VqmA transcriptional factors that has very important biological implications. In the present manuscript, the authors attempt to reveal the molecular mechanism of this transcriptional asymmetry by generating a battery of mutants in both VqmA receptors (bacterial and phage) and analyzing their ability to bind to the VqmR (bacterial) and qtip (phage) promoters by EMSA or with reporter gene, as well as to induce the lytic cycle of phage VP822. These assays evaluate the contribution of some residues and/or protein fragments to the transcriptional asymmetry in order to reveal their molecular determinants. The work is very well executed, the results are clear, the manuscript is well written and the logic is easy to follow, as is usual for the manuscripts of this group. However, I consider that it fails in its main claim, since the results do not allow the authors to define the molecular mechanism of the asymmetry. The results show that certain residues contribute to this asymmetry and that it possibly required the synergistic contribution of different portions or elements of the receptor to confer this asymmetry, but these data are not sufficient to propose a molecular mechanism. In fact, several alternatives are proposed in the discussion, some of which may be more feasible, but in no case can the data presented clearly answer the main question, the molecular mechanism of VqmA asymmetry, As is claimed at different points throughout the manuscript (e.g. pag 3, line 50: ”we discover the molecular mechanism underpinning the asymmetric transcriptional preferences of the phage-encoded and bacteria-encoded VqmA proteins). In this sense, I consider that the authors do not address this problem by making use of all available information, much of it generated by themselves, on the molecular mechanisms of the recognition of operators by VqmA. For VqmA-Vc, the molecular basis of VqmR operator recognition is quite well known since the three-dimensional structures of this protein bound to DPO (PDBs

6KJU and 6UGL, the last one solved by the manuscript’s authors) and, more importantly, VqmR operator (PDB 6IDE) have been reported. This last structure shows that VqmA adopts a quasi-symmetric dimeric form to recognize a GG palindromic sequence (GGN6CC) separated by 6 nucleotides (Wu et al, JBC 2019; PMID: 30610119). The direct readout of this sequence is performed by only 3 VqmA-Vc residues where a Lys (K185) has the major contribution. In fact, Bassler’s group showed in other manuscripts (Huang et al, JBC 2020; PMID: 31964715) that mutation of this Lys to Ala completely abolished the VqmA-Vc binding to its target DNA. In the present manuscript the authors show that the qtip promoter does not retain the GGN6CC box since the second CC has been replaced by TA, but the two nucleotides following this TA are CC, so that the GG palindrome could be regenerated again, but separated now by 8 nucleotides (GGN8CC) in its promotor. The authors show in this manuscript that the exchange of the region between the GG palindrome in the VqmR and qtip operators is sufficient to change the asymmetry (first section of results “VqmA promoter-binding selectivity is conferred by the DNA-binding domain”) but, surprisingly, they do not elaborate further on this direction to explain the asymmetry. Clearly, this observation opens three options, i) VqmA-phage is able to read both GGN6CC and GGN6TA, although with different affinity, ii) VqmA-Vc reads GGN6CC and VqmA-phage reads GGN8CC, and iii) VqmA-phage has a higher plasticity than VqmA-Vc and can recognize GG palindrome with 6 and 8 nucleotide separation. Although I see the second option as more likely, I think the authors should have tested three possibilities if they want to determine the molecular basis of the asymmetry.

Options 1 and 2 are easily analyzable and in fact the authors have already developed some of the tools to do it. The sequence alignments show that the Lys185 position at VqmA-Vc is occupied by an Arg in VqmA-phage, although both residues are similar in nature it is possible that this change may allow VqmA-phage to be more promiscuous and recognize both GGN6CC and GGN6TA or read alternatively GG palindrome separated by N6 o N8. The evaluation of this position should be a key point for understand the asymmetry. The second option is also easy to evaluate by carried out identical assays to those presented in the manuscript (EMSA and reported) where the CC pair in qtip operator is mutated. Finally, protein plasticity is harder to analyze but the authors can exploit and/or re-analyze the results presented in the manuscript using as guide the three-dimensional structures available to observe whether some of the mutations they find could confer more plasticity or alternatively induce a greater separation of the DBD at VqmA-phage than would be required to read GGN8CC. This could explain the effect observed for mutations in residues in the 201-211 region. It should be noted that the structure of VqmA-phage has recently been published (PDB 7DWM; Gu et al, BBRC 2021, PMID: 33895551). Although this structure should not have been available when the paper was written, I believe it may now help to understand the asymmetry.

Overall, I consider that the manuscript does not meet the requirements of PLoS Genetics (“substantial evidence for its conclusions”) as the data presented are not able to determine the molecular basis of the asymmetry observed in the VqmA receptor and therefore, unfortunately, I cannot support its publication.

**Have all data underlying the figures and results presented in the manuscript been provided?**

Reviewer #1: Yes

Reviewer #2: Yes

Reviewer #3: Yes

PLOS authors have the option to publish the peer review history of their article (what does this mean?). If published, this will include your full peer review and any attached files.

Reviewer #1: No

Reviewer #2: No

Reviewer #3: No

---

## [Decision Letter · Decision Letter 1]

16 Jun 2021

Dear Dr Bassler,

We are pleased to inform you that your manuscript entitled "Mechanism underlying the DNA-binding preferences of the Vibrio cholerae and vibriophage VP882 VqmA quorum-sensing receptors" has been editorially accepted for publication in PLOS Genetics. Congratulations!

Note that reviewers 2 and 3 have a few additional comments that you may wish to consider as you complete formatting changes prior to submission to production.

Yours sincerely,

Sean Crosson

Associate Editor

PLOS Genetics

Lotte Søgaard-Andersen

Section Editor: Prokaryotic Genetics

PLOS Genetics

Comments from the reviewers (if applicable):

Reviewer's Responses to Questions

**Comments to the Authors:**

Reviewer #1: The authors have addressed all my major and minor comments. Importantly, they have incorporated the structure recently published by Gu et al. into their discussion. I have no further comments, and I commend the authors on their excellent work.

Reviewer #2: The revised version of this manuscript much more clearly outlines the logic used throughout. Most importantly, the interpretations allowed by comparison of the results here to the newly published crystal structure as well as identification of the more widely spaced palindrome, while not answering all questions, makes this a much more complete story.

Minor suggestion:

I would suggest being a bit less definitive in ruling out HNS effects (no evidence of competition seen, but "eliminate the possibility"? How clear is it that the concentrations of everything are just what they are in vivo? RNA polymerase plus VqmA may displace better than VqmA alone?

Reviewer #3: The paper has been improved by the response to the reviewers and by the discussion of their results in parallel with the publication of Gu and coworkers. However, I cannot agree with the authors that they have proposed a mechanism that explains the asymmetry in recognition between both receptors, beyond the differences observed in the DNA operators (which both manuscripts confirm). From the receptors side, the conclusions of this asymmetry are restricted to "differences in amino acids between residues 171-224" corresponding to 50% of the DNA binding domain, which is rather poor to support what is claimed in the title "Mechanism underlying the DNA-binding differences.....". In any case, the enthusiasm shown by the other reviewers seems to indicate that my point of view must be pessimistic and inappropriate and perhaps the manuscript will serve as a point of support for further studies to reveal the molecular mechanism of this asymmetry as the authors indicate in the discussion "Going forward, determining the structure of VqmAPhage bound to Pqtip DNA should reveal the mechanism enabling recognition of Pqtip and the role that these residues play, individually and collectively, in driving DNA-binding specificity".

Undoubtedly the data presented in the manuscript support the conclusions expressed even though (in my opinion) not the title, therefore I have no problem in accepting it for publication if the editor judges it appropriate.

**Have all data underlying the figures and results presented in the manuscript been provided?**

Reviewer #1: Yes

Reviewer #2: Yes

Reviewer #3: Yes

PLOS authors have the option to publish the peer review history of their article (what does this mean?). If published, this will include your full peer review and any attached files.

Reviewer #1: No

Reviewer #2: No

Reviewer #3: No

**Data Deposition**

http://datadryad.org/submit?journalID=pgenetics&manu=PGENETICS-D-21-00489R1

**Press Queries**

---

## [Editor Report · Acceptance letter]

29 Jun 2021

PGENETICS-D-21-00489R1 

Mechanism underlying the DNA-binding preferences of the Vibrio cholerae and vibriophage VP882 VqmA quorum-sensing receptors 

Dear Dr Bassler, 

We are pleased to inform you that your manuscript entitled "Mechanism underlying the DNA-binding preferences of the Vibrio cholerae and vibriophage VP882 VqmA quorum-sensing receptors" has been formally accepted for publication in PLOS Genetics! Your manuscript is now with our production department and you will be notified of the publication date in due course.

With kind regards,

Katalin Szabo

PLOS Genetics

On behalf of:
